# Animal Models of Autosomal Recessive Parkinsonism

**DOI:** 10.3390/biomedicines9070812

**Published:** 2021-07-13

**Authors:** Guendalina Bastioli, Maria Regoni, Federico Cazzaniga, Chiara Maria Giulia De Luca, Edoardo Bistaffa, Letizia Zanetti, Fabio Moda, Flavia Valtorta, Jenny Sassone

**Affiliations:** 1Division of Neuroscience, San Raffaele Scientific Institute, 20132 Milan, Italy; bastioli.guendalina@hsr.it (G.B.); regoni.maria@hsr.it (M.R.); zanetti.letizia@hsr.it (L.Z.); valtorta.flavia@hsr.it (F.V.); 2Faculty of Medicine and Surgery, Vita-Salute San Raffaele University, 20132 Milan, Italy; 3Division of Neurology 5 and Neuropathology, Fondazione IRCCS Istituto Neurologico Carlo Besta, 20133 Milan, Italy; federico.cazzaniga@istituto-besta.it (F.C.); chiara.deluca@istituto-besta.it (C.M.G.D.L.); edoardo.bistaffa@istituto-besta.it (E.B.); fabio.moda@istituto-besta.it (F.M.); 4Laboratory of Prion Biology, Department of Neuroscience, Scuola Internazionale Superiore di Studi Avanzati, 34136 Trieste, Italy

**Keywords:** Parkinson’s disease, animal model, autosomal recessive Parkinsonism, dopaminergic neurons

## Abstract

Parkinson’s disease (PD) is the most common neurodegenerative movement disorder. The neuropathological hallmark of the disease is the loss of dopamine neurons of the substantia nigra pars compacta. The clinical manifestations of PD are bradykinesia, rigidity, resting tremors and postural instability. PD patients often display non-motor symptoms such as depression, anxiety, weakness, sleep disturbances and cognitive disorders. Although, in 90% of cases, PD has a sporadic onset of unknown etiology, highly penetrant rare genetic mutations in many genes have been linked with typical familial PD. Understanding the mechanisms behind the DA neuron death in these Mendelian forms may help to illuminate the pathogenesis of DA neuron degeneration in the more common forms of PD. A key step in the identification of the molecular pathways underlying DA neuron death, and in the development of therapeutic strategies, is the creation and characterization of animal models that faithfully recapitulate the human disease. In this review, we outline the current status of PD modeling using mouse, rat and non-mammalian models, focusing on animal models for autosomal recessive PD.

## 1. Introduction

Parkinson’s disease (PD) is the most common neurodegenerative movement disorder, affecting more than 6.1 million people worldwide [1]. It is characterized by progressive dysfunction and the death of dopamine (DA) neurons of the substantia nigra pars compacta (SNc). The ensuing DA depletion in the striatum causes motor symptoms (e.g., bradykinesia, rigidity, resting tremor, postural instability) and non-motor symptoms (e.g., depression, anxiety, weakness, sleep disturbances, cognitive disorders) [2,3]. Most often (90% of the cases), PD has a sporadic onset of unknown etiology possibly caused by the association of genetic and environmental risk factors. In 5–10% of cases, however, PD is monogenic with Mendelian inheritance [4]. At least 19 genetic loci for Parkinsonism have been identified to date, ten of which are autosomal dominant genes: *SNCA* (*PARK1*/*PARK4*), *PARK3*, *UCHL1* (*PARK5*), *LRRK2* (*PARK8*), *GIGYF2* (*PARK11*), *HTRA2* (*PARK13*), *VPS35* (*PARK17*), *EIF4G1* (*PARK18*), *TMEM230* (*PARK21*), and *CHCHD2* (*PARK22*). Autosomal-recessive (AR) homozygous or compound heterozygous mutations have been identified in nine genes: *PARKIN* (*PARK2*), *PINK1* (*PARK6*), *DJ-1* (*PARK7*), *ATP13A2* (*PARK9*), *PLA2G6* (*PARK14*), *FBXO7* (*PARK15*), *DNAJC6* (*PARK19*), *SYNJ1* (*PARK20*) and *VPS13C* (*PARK23*) [4] (Figure 1). Understanding the mechanisms behind DA neuron death in each of these Mendelian forms may help to illuminate the pathogenesis of DA neuron degeneration in the more common forms of PD.

A key step in the identification of the molecular pathways underlying DA neuron death, and in the development of therapeutic strategies, is the creation and characterization of animal models that faithfully recapitulate human disease. Ideally, an animal model for PD will display a progressive loss of SNc DA neurons and a major reduction of DA in the striatum. The models should also manifest motor symptoms (e.g., bradykinesia, rigidity, postural instability, resting tremor) and responsivity to L-DOPA. Widely employed in biomedical research because of their many advantages, mice and rats share biological similarity with humans (Chia 2020). Rodent models of PD can be broadly divided into environmental models, such as the widely used 1-methyl-4-phenyl-1,2,3,6-tetrapyridine (MPTP) mouse and the 6-hydroxydopamine (6-OHDA) rat model, and genetic models with knock-in (KI) and knock-out (KO) rodents based on known PD-associated genes [5]. Each model has its own strengths and limitations which determine the suitability of the model for a specific experiment [6].

Non-mammalian models are also extensively used to model PD. Among teleost fishes, zebrafish (*Danio rerio*) provides an interesting vertebrate model for the study of movement disorders because the neuronal circuitries involved in movement in zebrafish are well characterized, and most of the associated molecular mechanisms are highly conserved [7], demonstrating similarities with the mammalian central nervous system (CNS). Additionally, the DA neurons (14 in total) in the posterior tuberculum of the zebrafish (homologous to the substantia nigra in humans) are well characterized [7]. Clusters of DA neurons in the ventral diencephalon have ascending projections to the subpallium of the telencephalon, suggesting that these neurons are homologous to the ascending midbrain DA neurons of the mammalian nigrostriatal pathway [8]. Their relatively small size, optical transparency, rapid lifecycle and genetic similarity to humans make zebrafish a simple model for the evaluation of the pathological mechanisms in PD [7]. Several transgenic, knock-down (KD) and mutant zebrafish models of PD have been generated and characterized, advancing our understanding of the role of several genes implicated in the disease. Furthermore, the zebrafish vertebrate model is particularly suited for large-scale drug screening [6,7,9].

Although the zebrafish is the most widely used fish model, the Japanese Medaka teleost fish, *Oryzias latipes*, is also extensively used to model PD. Similarly to the zebrafish, Medaka fish are relatively transparent, produce large numbers of progeny per generation, are easy to maintain, and there are well-established techniques for the manipulation of their genomes [10].

Furthermore, *Drosophila melanogaster* provides a simple, yet powerful in vivo system to model PD pathobiology [11]. Clusters of dopaminergic neurons are detectable in the developing and adult fly, and metabolic pathways for DA synthesis are conserved between *Drosophila* and humans [9]. In the adult brain, *Drosophila* has distinct DA neuronal clusters, including about 200 DA neurons, and displays complicated behaviors mimicking some DA-dependent human behaviors [11]. *Drosophila* are cheap and easy to maintain in the laboratory; they have a rather short life span (40–120 days), and a variety of techniques and tools to manipulate their gene expression are currently available [11].

Other small organisms, such as *Caenorhabditis* (*C.*) *elegans*, are good experimental models to study PD because of their easy genetic manipulation and rapid reproduction and growth rate. *C. elegans* is a 959-cell nematomorph with a well-characterized genome. It exists primarily as a self-fertilizing hermaphrodite, in which the progeny are genetically identical. Males exist as a small fraction of the population (<0.1%) [12]. *C. elegans* has a well-defined, simple nervous system comprising 302 neurons and a DA network formed by eight neurons exactly: six anterior (four cephalic sesilla (CEP) neurons, two anterior deirids (ADE) neurons) and two posterior deirids (PDE) neurons. Male *C. elegans* possess six additional DA neurons in the tail ray [12]. Unlike other species in which staining for DA neurons is possible only in fixed tissues, neuronal survival in *C. elegans* can be assessed in live animals by expressing a fluorescent protein specifically in DA neurons, thanks to the existence of a transparent cuticle [12]. The quantification of more subtle phenotypes besides the loss of neuron cell bodies can be achieved by documenting the disappearance of axons, broken neurites, the retreat of dendritic terminals, and axonal and dendritic blebbing [12]. Here, we review the current state of animal models for autosomal recessive PD (ARPD) and discuss their limitations and utility.

## 2. *PARK2*: The Parkin RBR E3 Ubiquitin Protein Ligase Gene (*PARKIN*)

*PARK2-Parkin* (*PRKN*, OMIM 600116) was the first gene to be discovered to be linked to genetic Parkinsonism with an AR mode of inheritance [13,14]. Mutations in the *PARK2* gene are the most common known factor responsible for autosomal recessive juvenile Parkinsonism (ARJP) (10–20%) [15]. *PARK2* spans a large genomic interval (1.4 Mb) and is located in a region of genomic instability. The majority of parkin-proven PD cases result from large genomic alterations (deletion, duplication or inversion) that affect one or more exons. There are several other mutations, including missense and stop point mutations, but at a lower frequency than deletion [16,17]. *PARK2* encodes the protein PARKIN, a ubiquitin E3 ligase that, at the intracellular level, catalyzes the transfer of ubiquitin to various protein substrates. Through ubiquitination, PARKIN regulates the turnover and localization of many proteins, thus modulating a variety of cellular processes, including mitochondrial turnover [18,19,20]. Several transgenic overexpression and KO models of *PARK2* have been generated to gain insight into wild type (WT) PARKIN function and the mechanism of the disease [21] (Table 1).

### 2.1. Mouse Model

Several PARKIN mouse models have been generated, seven of which were Parkin KO models. In 2003, Itier and coauthors created the first Parkin KO mouse model by replacing murine Park2 exon 3 with the PGK-NeoR cassette (B6;129S2-Park2tm1Roo) [22]. While these mutant mice were viable and fertile, with normal brain morphology, weight and size, their bodyweight and body temperature were characteristically low. No overt behavioral change was observed up to 24 months of age. Furthermore, histological sections of the brain at the level of the striatum, hippocampus, brain stem and cerebellum revealed no differences versus WT animals, nor were differences in tyrosine hydroxylase (TH) staining observed in the SNc and the striatum. The immunohistochemical examination of the dopamine transporter (DAT) showed no significant differences between the WT and mutant mice. However, the levels of DAT and vesicular monoamine transporter type-2 (VMAT2) were significantly reduced in the striatum of the Parkin KO mice at 15 months of age. Behavioral tests showed that the Parkin KO mice displayed a deficit in learning and memory. Hence, this model did not display a clear PD phenotype. This led to the hypothesis that the nigrostriatal DA system in mice is less vulnerable to the neuronal damage induced by PARKIN absence than the human brain [22].

That same year, Goldberg and colleagues developed another *Parkin* KO mouse model carrying an exon 3 deletion (B6;129S4-Park2tm1Shn) [23]. They reported that the *Parkin* KO mice were viable, fertile, and exhibited normal brain morphology. Quantitative in vivo microdialysis revealed an increase in the extracellular DA concentration in the striatum. Intracellular recordings of medium-sized striatal spiny neurons showed a reduction in their synaptic excitability; in addition, behavioral tasks disclosed poorer performance on tests sensitive to nigrostriatal dysfunction [23]. Palacino and coauthors used this model to investigate whether the loss of PARKIN function resulted in an abnormal abundance and/or modification of proteins in the ventral midbrain [24]. Two-dimensional gel electrophoresis followed by mass spectrometry revealed a decrease in several proteins involved in mitochondrial function or oxidative stress [24].

A further exon 3-deleted *Parkin* mutant was generated by Stichel et al. in 2007 following the complete replacement of exon 3 with a NeoR cassette [25]. They observed severe mitochondria changes in the neuronal somata in the SNc. The mitochondria showed electron-dense inclusion bodies and dilated, disorganized cristae. These alterations coincided with a reduced complex I capacity in the SNc. Nonetheless, neither SNc DA neuron neurodegeneration nor motor disabilities were observed [25].

In 2005, Perez and Palmiter created a new *Park2* KO model involving the complete replacement of exon 2 with the Polr2a-NeoR cassette (B6;129S4-Park2tm1Rpa) [26]. The mice displayed no differences in their behavioral phenotype or in their catecholamine levels, suggesting that the nigrostriatal and the noradrenergic system were conserved [26]. Sato et al. created another exon 2-deleted *Park2* KO mouse model via the partial replacement of exon 2 with the MC1-NeoR cassette, followed by a frame insertion of tau-green fluorescent protein (tau-GFP) fusion protein [27]. This *Park2* KO mouse showed no obvious behavioral or locomotor activity deficits, no change in TH-positive nigra neurons, and no major decrease in DAT. The abnormally low levels of DA release were interpreted by the authors as a predictive factor for neuronal death [27]. Von Coelln and coauthors created another *Park2* KO model by deleting *Parkin* exon 7 (B6;129S7/S4-Park2tm1Tmd) [28]. The germline exon 7-deleted mice showed the early loss of catecholaminergic neurons in the locus coeruleus (LC) that mirrored the loss of LC neurons, as occurs in patients with *PARK2* mutations [47]. In addition to the impairment of the central noradrenergic system, there was a dramatic reduction in the norepinephrine-dependent acoustic startle response. A lower concentration of norepinephrine was also found in the olfactory bulb and in the spinal cord, two major target regions of projecting axons from the LC. There was no impairment of the nigrostriatal dopaminergic system [28]. The reasons behind the absence of SNc neuron degeneration in these *Park2* KO models remain to be elucidated. One plausible explanation is that genetic compensation occurs in response to the gene knockout [48].

In order to overcome any potential compensatory mechanisms that may underlie the absence of SNc DA neuron degeneration, Shin et al. followed a different strategy [29]. Because previous studies showed that the embryonic deletion of key factors often has no deleterious effects while the conditional KO of the same factor in adult animals leads to mice with profound neuron degeneration [49], Shin et al. generated a new conditional *Parkin* exon 7 KO mouse using the lentiviral delivery of GFP-tagged Cre-recombinase delivered stereotaxically to the midbrain of adult mice. Following this strategy, a progressive loss of DA neurons could be observed. Increased levels of PARIS (parkin interacting substrate) were also reported, as observed in both ARJP and sporadic PD human SNc [29]. Furthermore, Stevens and coauthors showed that the deletion of PARKIN in adult mice led to a decrease in mitochondrial size, mitochondria number, and protein markers in the ventral midbrain, consistent with a defect in mitochondrial biogenesis [30]. Hence, this conditional KO strategy seems to be promising.

A completely different model is the spontaneous quaking^viable^ mouse mutant. The genetic lesion in this model is a large deletion of approximately one megabase on chromosome 17 [31]. This deletion includes the *GkI* gene, which encodes the RNA-binding proteins of the STAR family [32]. The lack of GKL protein expression in oligodendrocytes leads to the improper myelination of the CNS in animals homozygous for the deletion. The quaking^viable^ mouse has been extensively studied for the dysmyelination of the CNS. The analysis of the sequence of the deleted region revealed that because the mouse homolog of the human gene *PARKIN* is partially contained in this interval, this deletion eliminates the expression of the mouse *Parkin* gene product and generates a mouse *Park2* KO [31]. Quaking^viable^ mice displayed normal cellular conformation in the grey matter. Although they have not been extensively studied, alterations in the DA metabolism have been reported in the quaking^viable^ mouse. Quaking^viable^ homozygotes showed reduced locomotor/exploratory activity, particularly in the open field test, and tremors in the caudal part of the trunk and proximal portions of the hind extremities. However, because this model did not show the loss of DA neurons in the SNc, the quaking^viable^ mouse cannot be considered a reliable model of PD [31]. The predominant dysmyelination phenotype further complicates behavioral, neuropathological and biochemical analysis [32].

In 2010, Ramsey and Giasson first observed a novel spontaneous missense mutation in the C3H mouse strain, resulting in PARKIN amino acid substitution E398Q [50]. Results from in vitro studies suggested that the novel E398Q *Parkin* mutant had reduced solubility and functional impairment [50]. This mutation is equivalent to an E399Q mutation in human *PARKIN*, because mouse *Parkin* has one residue fewer (Gly 139) than human *PARKIN* [51]. Because the E399Q variant has never been found in human subjects, it is not clear whether this variant can be considered a benign variant or a pathogenic mutation. 

In order to test the hypothesis that *Parkin* mutants may have toxic effects on DA neurons, Lu and colleagues generated a bacterial artificial chromosome (BAC) transgenic mouse expressing the truncated human parkinQ311X mutation in DA neurons, driven by a DAT promoter [33]. The private mutation Q311X is a C-to-T transition, replacing a Gln codon (CAG) with a stop codon (TAG) at amino acid position 311 in exon 8 [14]. ParkinQ311X mice have normal WT *Parkin* alleles on both chromosomes, and they express WT endogenous *Parkin* in addition to the exogenous human parkinQ311X variant ectopically expressed from the BAC chromosome. ParkinQ311X mice exhibit multiple late-onset and progressive hypokinetic motor deficits. Stereological analyses revealed that the mutant mice develop age-dependent DA neuron degeneration in the SNc accompanied by a marked loss of DA neuron terminals in the striatum. Neurochemical analyses showed a reduction in the striatal DA in mutant mice, which correlated with their hypokinetic motor deficits. Finally, mutant parkinQ311X mice exhibited the age-dependent accumulation of proteinase K-resistant endogenous α-syn in the SNc and increased levels of nitrotyrosine, a marker for oxidative protein damage [33]. This model was recently used to test a pharmacological neuroprotective strategy [52]. Although the genetics of this model do not exactly recapitulate AR transmission, it can be a useful model to study the deleterious effects of PARKIN mutants.

### 2.2. Rat Model

Recent advances in technology for mammalian genome engineering and the optimization of viral expression vectors have increased the use of genetic rat models of PD [53]. Compared to mice, some genetic rat models of PD better reproduce key aspects of PD; the reasons for the different phenotypes between mice and rats are not clear. Many behavioral, physiological and biochemical differences exist between mice and rats, in addition to the substantial differences in the gene sequence and expression [53]. Compared to mice, the rat neuronal circuitry more closely resembles that of humans; furthermore, rats are less prone to anxiety, which affords a major advantage for behavioral evaluation [9].

The *Parkin* KO rat model was generated using the zinc finger nuclease (ZFN)-mediated targeted disruption of exon 4 of the *Parkin* gene [34]. The rats displayed unaltered core body temperature and increased body weight, normal behavior, and no neurochemical changes [34]. A further characterization by Gemechu et al. showed that 2-month-old *Parkin* KO rats displayed lower monoamine oxidase (MAO) enzyme activity and higher levels of ß-phenylenthylamine (ß-PEA) than their WT counterparts [35]. *Parkin* KO rats displayed normal locomotor activity but a decreased locomotor response when tested with a low dose of psychostimulant methamphetamine, suggesting altered DA neurotransmission in the striatum when challenged with an indirect agonist [35]. Stauch and coauthors isolated non-synaptic mitochondria from the striatum of *Parkin* KO rats and analyzed the mitochondrial proteome: 15 proteins exhibited differential expression in *Parkin* KO rats [36]. This result suggested a pathological alteration; however, no mitochondrial dysfunction was reported [36]. As regards neuroanatomical changes in the DA neuron number, *Parkin* KO rats did not differ from WT rats, except for a small and not statistically significant 20% reduction in the DA neuron number at 8 months of age [34]. However, the *Parkin* rats were not analyzed at an older age. Given this small reduction in DA neurons, it is possible that in *Parkin* KO rats a potential PD-like phenotype is delayed, warranting further characterization in older rats [34,36]. Overall, this model can be useful to recapitulate some aspects of ARJP, but more data are necessary.

Building on previous work that suggested a dominant negative effect of PARKIN mutants [54], Van Rompuy and coauthors used a recombinant type 2 adeno-associated viral vector pseudotyped with type 8 capsid (rAAV2/8) to develop a novel rat model that overexpressed the human PARKIN-T240R mutation in the DA neurons of adult animals [37]. Surprisingly, the overexpression of PARKIN-T240R and WT PARKIN induced progressive DA neuron death in rats, starting at 8 weeks after viral injection [37]. The degeneration was specific for PARKIN because the similar overexpression of GFP did not lead to nigral degeneration. Nonspecific cell death induced by an inflammatory response to the vector preparations was excluded by immunohistochemical staining specific for activated and phagocytic microglia. This result contrasts with the many studies describing the neuroprotective capacity of PARKIN. The authors speculated that there might be a small window of physiologic PARKIN levels in DA neurons. Further data are needed to better characterize this model.

### 2.3. Zebrafish (Danio rerio)

Zebrafish Parkin protein is 62% identical to its human counterpart, with 78% identity in functionally relevant regions. The *park2* gene is expressed throughout zebrafish development and ubiquitously in adult zebrafish tissues [55]. Flinn et al. generated a zebrafish model of *parkin* KO using anti-sense oligonucleotides [38]. *Parkin* silencing led to a marked decrease in the number of ascending DA neurons in the posterior tuberculum (homologous to the substantia nigra in humans), which was enhanced by exposure to the toxin 1-methyl-4-phenylpyridinium (MPP^+^). Other neuronal populations were spared. Neither serotonergic nor motor neurons were affected; therefore, the toxic effect of *parkin* silencing was specific for DA neurons. Notably, *parkin* silencing impaired the mitochondrial function. Because this vertebrate model shared pathogenic mechanisms (reactive oxygen species (ROS), complex I deficiency) and a pathological hallmark (DA cell loss) with human *PARKIN*-mutated patients, the authors concluded that it mirrors the human disease better than any other currently available animal model for PARKIN disease [38]. Taniguchi et al. identified, in a library of fishes mutagenized by *N*-ethyl-*N*-nitrosourea (ENU), a nonsense mutation that resulted in a truncated Parkin protein at Tyr314. No further information is available regarding this model [56].

### 2.4. Drosophila melanogaster

A *Drosophila* model of ARJP was first created using a mutational approach to inactivate the highly conserved *parkin* ortholog. Flies bearing null alleles of *parkin* were viable but exhibited a reduced lifespan and male sterility [39]. The histological analysis of the major flight muscles revealed the severe disruption of muscle integrity, which correlated with locomotion defects [40]. Progressive mitochondrial pathology preceded myofibril degeneration; the muscle cells showed many swollen mitochondria with degenerated cristae. The standard histologic analysis of the brain revealed the appropriate development of the major brain centers in young and old mutant flies. DA neurons of the dorsomedial (DM), dorsolateral (DL) and anteromedial (AM) clusters of the medulla were analyzed but no clear neuronal loss was observed in any of these cell groups [40]. Lacking DA neurodegeneration, this model failed to recapitulate the prominent hallmark of PD.

The observed mitochondrial dysfunction was then investigated in order to understand the mitochondrial impairment occurring in DA neurons. One year later, Pesah et al. used P-element mutagenesis to generate another *parkin* KO mutant, and they confirmed previous findings by Greene et al. (2003). They observed female infertility and increased sensitivity to chemical and environmental stress in mutant animals compared to controls [41]. Again, no major loss of DA neurons in the *parkin* mutant was observed. In order to better understand the effects of *parkin* mutations in vivo, Wang and colleagues generated transgenic *Drosophila* overexpressing the human *PARKIN* mutant R275W associated with ARJP [42]. Transgenic flies overexpressing R275W displayed an age-dependent degeneration of specific DA neuronal clusters and concomitant locomotor deficits that accelerated with age or in response to rotenone treatment. The R275W mutant flies exhibited prominent mitochondrial abnormalities in their flight muscles. These defects were caused by the expression of human parkinR275W and were highly similar to those observed in *parkin* null flies.

These findings raised the interesting hypothesis that selected *parkin* mutations may directly exert neurotoxicity in vivo [42]. Cha et al. generated two other transgenic lines in *Drosophila* expressing an N-terminal deleted *parkin* (containing only 108–482 aa) and *parkin*K71P, a point mutant similar to R42P found in ARJP patients [43]. Both *parkin* mutant flies showed reduced longevity, a drooped wing phenotype, locomotor dysfunction and muscle degeneration accompanied by apoptosis. Among the histopathological hallmarks, there was a severe loss of DA neurons in the *parkin* mutant flies and a shrunken morphology of DA neurons [43]. These phenotypes were completely restored by the overexpression of WT *parkin*. In order to further explore whether the *parkin* mutations identified in familial PD can exert cell-specific toxic effects in vivo, Sang and colleagues generated *Drosophila* lines expressing parkinQ311X or T240R mutants under the control of a ddc-GAL4 driver to lead specific Parkin expression in DA neurons and serotonin (5-HT) neurons [44]. Flies expressing parkinT240R or parkinQ311X showed a dramatic decline in their climbing ability compared to the controls. Motor tests revealed severe deficits with onset 2–3 weeks after eclosion, which suggested a degenerative, rather than a developmentally induced, defect. A histopathological hallmark was age-dependent selective DA neuron degeneration [44]. The model provided a robust behavioral and neuropathological phenotype.

### 2.5. Caenorhabditis elegans

*C. elegans* has orthologs to many of the genes implicated in PD (e.g., *LRRK2*/*lrk*-*1*, *PINK1*/*pink*-*1*, *PARKIN*/*pdr*-*1*, *DJ*-*1*/*djr*-*1.1*/*djr**-1.2*, *ATP13A2*/*catp*-*6*). This phylogenetic similarity to humans makes *C. elegans* a suitable PD model [12]. Ved and colleagues generated a *parkin* KO *C. elegans* line [45] that displayed normal development but had a shorter lifespan than the WT. The strain was also more vulnerable to several mitochondrial complex I inhibitors than nontransgenic nematodes [45]. Bornhorst et al. used a PARK2-*pdr1*(gk448) III (CGC) *C. elegans* strain to investigate manganese (Mn)-induced toxicity in dopaminergic neurotoxicity. The mutants exhibited hypersensitivity to Mn-induced lethality compared to WT worms and a time-dependent increase in Mn-induced reactive oxygen and nitrogen species (RONS). However, when the authors used worms expressing GFP under the control of a promoter for the dopamine re-uptake transporter 1 (*C. elegans* ortholog for vertebrate DAT), they observed in these mutants no Mn-induced degeneration of the CEP dopaminergic neurons [46].

## 3. *PARK6*: PTEN-Induced Putative Kinase 1 (PINK1)

Identified in 2004, mutations in the *PARK6* gene (*PINK1*, OMIM 605909) are the second most common genetic cause of autosomal recessive early-onset PD (AR EOPD) [57]. *PARK6* encodes a 581-amino-acid protein named (PTEN)-induced kinase 1 (PINK1) with an N-terminal mitochondrial targeting sequence (MTS) and a short transmembrane domain (TMD), followed by a C-terminal serine/threonine kinase catalytic domain [58]. PINK1 is imported into the mitochondria through the translocase of the outer (TOM) and inner (TIM23) membrane complexes; its MTS is cleaved off by the mitochondrial processing peptidase (MPP) located in the matrix [58]. The Pink1 TMD acts as a stop-transfer sequence leading to integration into the inner membrane, where PINK1 exposes its kinase domain to the mitochondrial intermembrane space (IMS) [59]. Subsequently, the inner mitochondrial membrane protease presenilin-associated rhomboid-like protease (PARL) cleaves PINK1 into a short soluble form which is then released into the cytosol [58,60]. The disruption of the mitochondrial membrane potential interferes with PINK1 mitochondrial import through the TIM23 complex and induces PINK1 retention and accumulation on the outer mitochondrial membrane. Here, PINK1 participates in inducing mitochondrial turnover [61,62,63].

PINK1 has been suggested to provide protection against oxidative stress by assisting in condemning damaged mitochondria to degradation/mitophagy and maintaining mitochondrial homeostasis [64]. PINK1 deficiency in mammalian neurons alters the mitochondrial buffering capacity, increases ROS and impairs respiration [65], suggesting a mechanism by which the loss of PINK1 function would confer vulnerability to cell death. Most PD-linked mutations are located within the kinase domain and result in impaired or the loss of function of kinase activity and the decreased neuroprotective function of PINK1 [66]. In order to investigate the physiological function of PINK1 and its potential neuroprotective role, several PINK1-deficient animal models have been generated (Table 2).

### 3.1. Mouse Model

In 2007, Zhou et al. used a transgenic RNA interference (RNAi) approach to generate a conditional *Pink1*-silenced mouse. In these transgenic mice, PINK1 expression was regulated by the Cre-LoxP system; the activation of the RNAi transgene resulted in the silencing of the *Pink1* gene to less than 5% of normal levels in the CNS [67]. The silencing of the *Pink1* gene expression did not cause a loss of the DA neurons in the SNc, an alteration in the DA level in the striatum, or abnormalities in motor function [67]. In addition, *Pink1* KO mouse models were generated and characterized [68,69]. The *Pink1* KO mice generated by germline deletions of exons 4–7 did not exhibit major abnormalities or PD-associated pathology in the brain. No changes in the number of DA neurons or striatal DA levels were observed, and no alterations in DA synthesis or DA receptor levels were found [69]. Nevertheless, the absence of PINK1 caused a marked decrease in the evoked DA release in striatal slices and a reduction in the quantal size and release frequency of catecholamines in dissociated chromaffin cells [69]. The impairment of DA release in *Pink1* KO mice compromised striatal synaptic plasticity, with a reduction in corticostriatal long-term potentiation (LTP) and long-term depression (LTD), which could be restored by either DA receptor agonists or agents that increase DA release, such as amphetamine or L-DOPA [69].

Further studies with the same mouse model showed that the loss of PINK1 also resulted in mitochondrial dysfunction [68]. While the absence of PINK1 did not cause gross ultrastructural alterations or changes in the total number of mitochondria, an increased number of larger mitochondria was observed in the striatum of *Pink1* KO mice at 3–4 and 24 months. Mitochondrial respiration was also impaired in young *Pink1* KO mice, and the activity of complex I, complex II and aconitase was reduced in the striatum but not in the cerebral cortex. At 2 years of age, mitochondrial respiration was also impaired in the cerebral cortex, indicating an age-dependent impairment of mitochondrial function [68]. The mitochondria in the cerebral cortex were found to be more sensitive to oxidative stress [68].

Glasl and colleagues generated another *Pink1*-deficient mouse by deleting exons 2 and 3; this excision generates a shift of the reading frame that results in a stop codon in exon 4 [70]. As observed in other *Pink1* KO mice [67,68,69], these mice did not show morphological alterations in the DA system. No differences in the number of TH-positive neurons in the SNc or in the striatal DA content were observed at 6 or 19 months of age. The neurons in the *Pink1* KO mice showed less fragmented mitochondria. Behavioral analysis in the open field showed that the mice, at the ages of 3 and 24 months, did not manifest a strong impairment in their spontaneous locomotor activity in a novel environment. The mice demonstrated normal motor coordination and balance [70]. Nonetheless, at 26 months of age the *Pink1*-deficient mice displayed symptoms resembling the early phases of PD, symptoms that in humans are known to precede the severe motor-phase of the disease: gait alterations, especially in the hind paws, and olfactory dysfunction. The density of the serotonergic fibers in the glomerular layer of the olfactory bulb was considerably reduced and the aged male *Pink1* KO mice showed a marked deficit in their fine olfactory discrimination and smell sensitivity, which are two non-motor symptoms of PD [70].

In line with these findings, Kelm-Nelson and colleagues used the *Pink1* KO mouse (*Pink1^tm1Shn^*) to investigate early-onset locomotor and sensorimotor deficits, including deficits in the limbs, and impaired vocalization, articulation and swallowing [71]. The *Pink1* KO mice exhibited impaired vocalization: they produced simple calls at a lower intensity compared to WT. The assessment of their spontaneous motor activity showed a severe impairment of limb motor skills, with fewer hindlimb and forelimb steps and reduced rearing and landing on the cylinder test compared to WT. Additionally, the *Pink1* KO mice took longer to turn and traverse during the pole test. These motor deficits occurred in the absence of DA neuron loss in the SNc or a reduction in TH optical density in the striatum [71]. While *Pink1* KO mice do not recapitulate DA neuron loss, they could be useful to study early dysfunctions associated with PINK1-related PD.

Gispert and colleagues generated a transgenic mouse bearing the human pathogenic G309D-*PINK1* missense mutation inserted by homologous recombination into the orthologous mouse *Pink1* locus. The G309D mutation is a loss-of-function mutation that induces *Pink1* mRNA instability and dampens PINK1 protein expression [72]. An analysis of *Pink1* mRNA showed a 97% reduction in the brain of homozygous mice. There was an age-dependent reduction in DA and a decrease in locomotor activity, with selectivity for spontaneous movements [72], along with a progressive reduction in body weight starting from middle age. No Lewy body formation or nigrostriatal degeneration was observed up to 18 months of age; however, impaired bioenergetics and progressive mitochondrial dysfunction in the brain were detected. There was a reduction in the mitochondrial membrane potential and adenosine triphosphate (ATP) levels, and respiratory complex activity was decreased [72]. Despite the reduction in fission factor *Mtp18*, the G309D-*Pink1* mice showed normal mitochondrial morphology and mass; only after proteasomal stress did they show reduced fission and increased perinuclear aggregation of the mitochondria, perhaps due to the failed activation of mitophagy [72].

Another *Pink1* KO mouse was generated by the deletion of a large portion of the gene, from exons 2 to 5, and primary neuronal cultures were prepared [73]. The major finding of that study conducted on in vitro cortical neurons was an age-dependent reduction in long-term viability and higher cytotoxicity indices compared to the control neurons, with an increase in cell death and apoptosis [73]. Using the same *Pink1* KO transgenic mouse, Gandhi and colleagues explored the role of PINK1 in maintaining mitochondrial function in in vitro neurons [65]. They reported that the loss of PINK1 function in primary cortical and midbrain neurons increased Ca^2+^ levels, causing mitochondrial Ca^2+^ overload and excess ROS production. PINK1 deficiency was also associated with reduced glucose uptake that resulted in the global impairment of respiration and the loss of mitochondrial membrane potential, which sensitized the mitochondria to the opening of the mitochondrial permeability transition pore (mPTP) [65].

### 3.2. Rat Model

*Pink1* KO rats were generated using ZFN technology [34,74]. In their preliminary characterization, Sun and colleagues performed a quantitative autoradiography analysis of DA pre-synaptic markers and post-synaptic receptors in the striatum of 6-month-old mice, and they found altered DA regulation. The density of DA D_2_ and D_3_ receptors was significantly increased in the striatum of transgenic rats, while no remarkable changes in the density of DA D_1_ receptor, VMAT2 or DAT were observed [74]. Later, Dave and colleagues performed a better phenotypic characterization of the model [34], showing that the loss of the *Pink1* gene produced robust behavioral dysfunction and a severe loss of SNc DA neurons. Motor impairment in movement, strength and coordination started at 4 months of age and preceded nigral neuronal loss. The loss of DA neurons was rapid between 4 and 8 months of age, with a decrease of about 25% seen at 6 months and 50% at 8 months of age [34]. The loss of midbrain DA neurons in this model was confirmed by a later study: the number of TH-positive neurons in the SNc was decreased, as well as the size of the SNc at 9 months of age [75]. While nigral neurodegeneration did not correlate with changes in the striatal TH or α-syn immunoreactivity within the SNc, striatum or other brain areas, there was a 2–3-fold increase in DA and 5-HT striatal content at 8 months of age [34]. *Pink1* KO rats also displayed metabolic alterations and mitochondrial dysfunction [75]; selected brain metabolomic markers produced in the mitochondria or interacting with the mitochondrial products were altered. In the striatum, aspartate was found to be increased while taurine and creatinine were decreased; myoinositol was found to be decreased in the cortex.

Regarding mitochondria, an analysis of the expression of electron transport chain (ETC) complex I, III, IV and V subunits revealed a decrease in the striatum of *Pink1* KO rats, while the complex II subunit levels were unchanged. Proton leak was increased in the brain mitochondria of *Pink1* KO rats at 4 and 9 months of age, suggesting increased ROS generation. This is consistent with the increased ROS signaling and altered stress pathway observed in the striatum of 4-month-old *Pink1* KO animals [75]. Furthermore, the striatal glycolytic protein levels were higher, suggesting increased glycolysis. Additionally, the mitochondrial trafficking and dynamics were altered in the striatum of *Pink1*-deficient mice at 4 months of age: dynamin related protein-1 (DRP1) was increased while mitofusin-2 (MFN2) was decreased; these results paralleled increased mitochondrial fission and fragmentation [75].

Grant and coauthors evaluated the effect of the loss of PINK1 function on vocalization, oromotor and limb function, and neurodegenerative features [76]. They showed that *Pink1*-deficient rats developed the early and progressive impairment of their vocalization: already, at 2 months of age, the rats produced calls which were lower in intensity and bandwidth compared to WT, while the peak frequency was decreased at 6 months of age. The loss of function of PINK1 also caused oromotor deficits, including variable licking behavior and irregular and inconsistent biting patterns. In contrast to the vocalization and licking deficits, which emerged early, sensorimotor deficits occurred at 8 months of age with a reduction in spontaneous activity and an increase in the time needed to travel a tapered beam. Though no differences in TH immunoreactivity were found in the striatum and in the SNc at 8 months of age, *Pink1* KO rats showed a reduction in TH immunoreactivity in the LC. Furthermore, insoluble α-syn aggregates were found in the SNc, in the periaqueductal gray (PAG) and in the LC of 8-month-old *Pink1* KO rats.

Pultorak and coauthors explored the consequences of the vocalization impairment observed in *Pink1* KO rats on social communication [76] as a measure of female approach behavior [77]. By presenting recordings of male *Pink1* KO and normal WT ultrasound vocalizations (USVs) to female rat listeners, the authors observed that the female rats were less motivated by the male *Pink1* KO USVs than the WT male USVs. Females exposed to *Pink1* KO USVs showed decreased neuronal activity in the *nucleus accumbens*, a region implicated in auditory processing and sexual motivation. The results demonstrated that vocalization deficits in a PINK1-PD related model compromised communication [77]. The studies showed that *Pink1* KO rats can be considered a promising model to study PINK1-associated PD.

### 3.3. Zebrafish (Danio rerio)

The zebrafish *pink1* gene (NM_001008628) is located on chromosome 23 and shares 53% identity with humans. It encodes a 574 aa protein which is 54% identical to the human PINK1 protein. An analysis of the zebrafish primary sequence revealed a putative mitochondrial targeting domain at the N-terminus of the protein and a putative serine/threonine protein kinase active site, which is highly similar to human PINK1 [78]. The model of *pink1* deficiency in zebrafish was created by using morpholino (MO) antisense; this model exhibits a severe developmental phenotype, including structural alterations in the axonal scaffold and a decrease in the number of central DA neurons. The phenotype was rescued by the overexpression of WT human *PINK1* mRNA [78]. MO KD of the zebrafish *pink1* ortholog added evidence for the importance of this gene in the control of oxidative stress and mitochondrial function. The downregulation of Pink1 expression resulted in mitochondrial dysfunction, with a reduction in the mitochondrial membrane potential, augmented ROS levels, and the activation of the apoptotic signaling pathway, including increased caspase-3 activity. Furthermore, *pink1* KD was associated with the elevated activity of the mitochondrial protein glycogen synthase kinase 3 β (GSK-3β), whereas its inhibition via nonspecific (LiCl) and specific (SB216763) inhibitors partially rescued the phenotypes in *pink1* morphant zebrafish [78]. 

Priyadarshini et al. found that the expression of 177 genes was altered in Pink1 morphants. Among the canonical pathways identified, the most affected was hypoxia-inducible factor (HIF) signaling with the strong downregulation of *hif1α* mRNA. Transforming Growth Factor-β (TGF-β) signaling, retinoic acid receptor activation, and the biogenesis of the mitochondria were also altered [79]. Catalase enzyme activity was decreased, coupled with *catalase* and *SOD3* transcript downregulation. These enzymes are part of the antioxidant defense system. Both catalase and superoxide dismutase are ROS-scavenging enzymes; consistently, their downregulation was associated with an increase in ROS levels. The Pink1 morphants showed an abnormally lower heart rate but no gross morphological phenotype. Following *pink1* KD, ferrochelatase expression was also upregulated, leading to the increased expression of vascular endothelial growth factor (VEGF) and erythropoiesis. The increased erythropoiesis effect was rescued by the administration of the antioxidants N-acetyl cysteine (NAC) and L-glutathione reduced (LGR), while the abnormalities of the gene expression profiles were rescued by *pink1* mRNA injection [79]. In another study, the MO-mediated KD of *pink1* function did not cause large alterations in the number of dopaminergic neurons in the ventral diencephalon, though the patterning of these neurons and their projections was perturbed [8]. This was accompanied by locomotor dysfunction, a weak or absent response to tactile stimuli, and reduced swimming behavior. These defects were rescued by the expression of exogenous Pink1 or the administration of the DA D_1_ receptor agonist SFK-38393 [8].

Another study reported no morphological or behavioral deficits in *pink1* morphant zebrafish [80]. However, an increased susceptibility to MPTP-induced motor disturbance was observed: the DA neuron clusters of *pink1*-deficient zebrafish were more sensitive to MPTP toxicity. In addition, the translation inhibition of Pink1 reduced the *th1* and *th2* mRNA forms without affecting the levels of *dat* mRNA, and this effect was rescued by the injection of *pink1* mRNA. Although there was a decrease in the number of TH-positive neurons in the DA diencephalic cluster, the normal DAT levels suggested that the downregulation of Pink1 may cause a decline in essential mRNAs and proteins (e.g., TH) in the absence of effects on neuronal survival [80]. The increased susceptibility to MPTP underlined the importance of Pink1 in oxidative stress: exposure to hydrogen peroxide (H_2_O_2_) dramatically upregulated the expression of *pink1* mRNA and decreased *th2* mRNA in zebrafish. Both effects could be reverted by treatment with the antioxidant LGR [93]. LGR could also rescue, along with *pink1* mRNA, *th1* and *th2* expression, which were lost in Pink1 morphants [80,93].

A *pink1* mutant zebrafish line (*pink1*^−/−^) with a premature stop mutation in exon 7 (Y431*) in the *pink1* homolog was found in ENU mutagenesis libraries [55]. This mutation resulted in a truncated Pink1 protein, with the loss of its C-terminus and part of its kinase domain, leading to the inactivation of Pink1 catalytic activity and decreased mRNA stability [81]. While these adult *pink1*^−/−^ fishes did not display obvious behavioral abnormalities, *pink1*^−/−^ embryos at 5 days postfertilization (dpf) showed a marked decrease in their number of DA neurons and a reduction in their mitochondrial complex I activity [55]. Another study confirmed that Pink1 deficiency resulted in a progressive loss of DA neurons from early development (5 dpf) to adulthood (18 months) [81]. *Pink1*^−/−^ embryos did not display any overt morphological abnormalities and the expression of several neurodevelopmental markers was unchanged compared to WT embryos. These results contrasted with findings by Anichtchik et al., who observed a severe developmental phenotype with major generalized neurodevelopmental abnormalities in the MO-mediated knock down of *pink1* in zebrafish embryos [78]. Despite the absence of overt morphological abnormalities, Pink1 deficiency resulted in severe alterations in mitochondrial function and morphology: a reduction in mitochondrial complex I and III activity and enlarged mitochondria were observed during early development and adulthood. The expression of *TigarB*, the zebrafish ortholog of the human glycolysis and apoptosis regulator TIGAR, was markedly increased in the *pink1* null mutant. The antisense-mediated inactivation of TigarB normalized the mitochondrial function. *Pink1* mutant larvae also displayed a marked increase in microglial activation, but the inactivation of the microglia failed to rescue the DA neuron loss, suggesting that microglial activation may be a downstream mechanism involved in DA neuron loss in Pink1 deficiency [81].

### 3.4. Medaka Fish (Oryzias latipes)

A single ortholog of the human *PINK1* gene was identified in the medaka genome: the medaka *pink1* gene has 8 exons and encodes a protein consisting of 577 amino acids which has 54.1% homology to human PINK1. The kinase domain is highly conserved between the two species.

Several genetic models were created using the targeting-induced local lesions in genomes (TILLING) method [82]. Matsui and coauthors generated a *pink1* mutant medaka fish by screening the TILLING library. Among the 14 mutations found in exons 2 and 3 of the *pink1* gene, the nonsense mutation Q178X, which resulted in the disruption of the kinase domain and the degradation of *pink1* mRNA, was selected to generate homozygous mutants (*pink1^Q178X^*^/*Q178X*^) [82]. The *pink1* mutant grew normally at first, without any obvious morphological abnormalities or developmental disorders; they showed a normal phenotype for germ-cell lineage, skeletal muscle and mitochondrial morphology. Then, in the late-adult stage, the frequency of their spontaneous swimming movements dramatically decreased and death occurred at 12 months, in a lifespan shorter than that of the control fishes. Although the loss of Pink1 caused the dysregulation of the DA metabolism with a decrease in the amount of 3,4-dihydroxyphenylacetic acid (DOPAC), the mutants showed no evident defect in the number or morphology of their DA neurons [82].

### 3.5. Drosophila melanogaster

The *Drosophila pink1* gene (CG4523) encodes a polypeptide of 721 amino acids (about 80 kDa in molecular mass). Similar to human Pink1, a structural analysis of the *Drosophila* Pink1 protein revealed an MTS and a serine/threonine kinase domain that shares 60% similarity and 42% amino acid identity with human PINK1. Consistent with the localization of human PINK1, *Drosophila* Pink1 was also found to localize in mitochondria [83].

Loss-of-function Pink1 mutants exhibited male sterility and mitochondrial defects (e.g., enlarged swollen mitochondria with the loss of the outer membrane and reduced ATP levels). The phenotype was restored by Pink1 expression. Furthermore, mitochondrial dysfunction led to the degeneration of flight muscles and the mild loss of DA neurons [83]. *Pink1* mutant flies shared marked phenotypic similarities with Parkin mutants: abnormally positioned wings, a crushed thorax, disorganized muscle fibers with enlarged mitochondria, muscle cell apoptosis, impaired flight ability, a severely reduced climbing rate and complete male sterility [39,40,41,43]. Clark and coauthors observed a similar phenotype after the removal of *pink1* in *Drosophila*: male sterility, apoptotic muscle degeneration, defects in the mitochondrial morphology (fragmented mitochondrial cristae) and a reduction in ATP, but no change in the number of DA neurons. *Pink1* mutants showed increased sensitivity to multiple stresses, including oxidative stress. These phenotypes were fully suppressed by *pink1* expression [84]. Morais and coauthors, using previously characterized *pink1* mutant flies [83,84], observed that *Drosophila pink1* mutant flies fail to maintain normal synaptic transmission and show the impaired mobilization of synaptic vesicles from the reserve pool during rapid stimulation [85]. This was caused by synaptic ATP depletion. The expression of human PINK1, but not its clinical mutant G309D, rescued the mobilization of reserve pool vesicles. The absence of Pink1 caused deficits in the mitochondrial membrane potential and defects in the catalytic activity of ETC complex I, but no alteration in mitochondrial morphology [85]. 

Another study reported that the *Drosophila pink1* KO model showed a severe reduction in mitochondrial respiration driven by the ETC in the mitochondria, resulting from a decrease in ETC complex I and IV enzymatic activity [86]; in addition, the assembly of ETC complex I was compromised, as previously reported [85]. As a consequence, the *pink1*-KO flies also displayed reduced mitochondrial ATP synthesis [86]. It was previously observed that DRP1, a key molecule in mitochondrial fission, can rescue mitochondrial fission and morphological abnormalities in *pink1*-KO flies [94,95]. Liu et al. observed that DRP1 expression restored the ATP synthesis rate, rescued complex I and IV activity deficits, and increased the amount of assembled ETC complex I in *pink1*-KO flies [86]. Yang and coauthors used the transgenic RNAi approach to knock down the expression of *pink1* in *Drosophila* [87]. The inhibition of Pink1 function in *Drosophila* severely reduced their lifespan. *Pink1* RNAi flies also showed defects in their wing posture, reduced climbing ability, and abolished flight capacity by 10 days of age. Their muscle pathology was characterized by the severe disruption of muscle integrity, followed by the degeneration of selected indirect flight muscles with extensive DNA fragmentation, preceded by mitochondrial dysfunction. The ATP level was reduced, the mitochondria were grossly swollen, lacked electron-dense material, and showed the disintegration of the cristae. The inactivation of Pink1 led to a degeneration of TH-positive neurons and a reduction in brain DA content [87].

### 3.6. Caenorhabditis elegans

The *C. elegans* ortholog of human *PINK1* was identified: it was named *pink**-1* and, like human PINK1, it consists of two characteristic motifs: an N-terminal MTS and a serine/threonine kinase domain that shares 36% identity and 54% similarity to human PINK1 [88]. Sämann et al. characterized a deletion mutant of *pink*-*1*. The *pink*-*1*(*tm1779*) mutants harbor a 350-bp deletion that eliminates part of the promoter region and the first two exons, including the proposed transcriptional and translational start sites, resulting in complete functional loss [88]. The authors demonstrated that the loss of *C. elegans pink**-1* resulted in increased paraquat sensitivity and decreased oxidative stress response. This dysfunction was accompanied by mitochondrial phenotypes characterized by a reduction in the mitochondrial cristae length (12% in muscle cells and more than 30% in neurons). Furthermore, the *pink*-*1*(*tm1779*) mutants displayed defects in the axonal outgrowth of the canal-associated (CAN) neurons, a pair of migratory neurons in the midbody region of the worm that are frequently used as an indicator of axonal guidance and cell migration defects [88].

In 2015, Luz et al. analyzed the mitochondrial morphology and the fundamental parameters of the mitochondrial respiratory chain in the *C. elegans pink*-*1*(*tm1779*) model [89]. No differences were observed in the basal oxygen consumption rate (OCR), but treatment with *N*,*N′*-dicyclohexylcarbodiimide (DCCD) caused a major reduction in OCR coupled to ATP consumption. Furthermore, exposure to the mitochondrial uncoupler carbonyl cyanide-p-trifluoromethoxyphenylhydrazone (FCCP) caused a marked increase in OCR above basal levels, while no differences were observed in the maximal or spare respiratory capacity. *Pink*-*1*-deficient nematodes showed severe proton leakage and a highly fused mitochondrial network that exhibited larger aspect ratios compared to the WT nematodes, which is consistent with the loss of mitophagy and a reduction in the mitochondrial turnover [89]. In line with these findings, elevated mitochondrial ROS generation, decreased ATP levels and decreased mitochondrial membrane potential, suggestive of proton leak, were recently reported in the same model [90]. The authors also found increased basal OCR and a fragmented and disorganized mitochondrial network morphology. They observed that in *pink*-*1*-deficient nematodes, mitophagy is not activated under stress, which leads to the accumulation of dysfunctional and damaged mitochondria and a shortened lifespan [90]. In agreement with these findings, *pink*-*1* silencing abolished the detectable removal of ultraviolet C (UVC)-induced damaged mitochondrial DNA (mtDNA) due to a lack of mitophagy [91,92].

## 4. *PARK7*: The Parkinsonism-Associated Deglycase Gene (*DJ*-*1*)

The *PARK7* or *DJ*-*1* gene (OMIM 606324), localized on chromosome 1p36.23, was discovered to cause AR EOPD in 2001 [96]. *DJ**-1* mutations are quite rare: the frequency is 0% to 1% in early-onset PD cohorts [97]. Molecular analyses for *DJ-1* mutations have identified homozygous and compound heterozygous mutations, including missense, truncating and splice-site mutations, and large deletions [4,98,99]. The DJ-1 protein is a redox-sensitive molecular chaperone protein, ubiquitously and highly expressed in brain areas and extra-cerebral tissues, localized in the cell cytosol, nucleus and mitochondria [100,101,102]. Some *DJ*-*1* mutations cause a loss-of-function of the DJ-1 protein, thus inducing the instability of the DJ-1 dimer, while other mutations induce a lack of DJ-1 expression [103,104]. Studies have shown that *DJ*-*1* mutations can generate ROS, leading to oxidative stress, mitochondrial dysfunction and cell death [4,105], but the precise mechanisms by which DJ-1 deficiency leads to PD remain elusive [4,105] (Table 3).

### 4.1. Mouse Model

The first *Dj-1* KO mouse model was generated by replacing exons 3–5 with a neomycin (Neo) selectable cassette. These *Dj-1* null mice were viable and fertile, and did not display any gross neuronal abnormalities or motor deficits observed on the pole, the open field or the adhesive removal test [106]. A stereological analysis of their SNc DA neurons showed no alterations in the number of TH-positive neurons; striatal immunostaining showed no change in the TH fiber density or DAT level. The quantification of the DA by High Performance Liquid Chromatography (HPLC) revealed no change in the striatal region [106]. However, the mice were more susceptible to MPTP-induced neuron loss, and the restoration of DJ-1 expression via adenoviral vector delivery mitigated this phenotype. The authors also showed that the overexpression of DJ-1 via an adenoviral vector in WT mice prevented MPTP-induced neuronal loss and protected the animals against neurodegeneration in the SNc [106]. Using the same mouse model, Zhou and coauthors reported that *Dj-1* null mice cannot perform the running wheel or the rotarod test at the same intensity as WT animals [107]. A second mouse model was generated by Chen and coauthors in 2005 by deleting 9.3-kb genomic DNA, including the first five exons and part of the promoter region of the *Dj-1* gene. The mice were healthy and fertile, and appeared indistinguishable from the WT at birth. No motor deficits were detected at any age on the rotarod test, whereas the open field test revealed impairment at 11 months of age, and the tape removal task disclosed nigrostriatal dysfunction at 5 and 11 months [103]. Voltammetry studies of these *Dj-1* null mice showed an increase in the evoked DA release in the dorsal striatum but not in the *nucleus accumbens* core and shell, and the HPLC dosage in the striatum displayed an age-dependent increase in the DA content [103]. Immunohistochemical analyses showed no differences in the TH, DAT or VMAT2 protein levels, no death of DA neurons, and no α-syn- or ubiquitin-positive inclusions were found in the SNc [103]. 

Another *Dj-1* KO mouse was generated by Goldberg et al. in 2005 by targeting *Dj-1* exon 2 [99]. Similarly to the other *Dj-1* mouse models, this one did not display DA neuron loss or a deficit in the DA levels in the basal ganglia. The mice had normal TH activity, as measured by L-DOPA quantification determined by HPLC in the presence of an inhibitor of L-DOPA decarboxylase, and they did not have α-syn- or ubiquitin-positive inclusions [99]. Unlike the other models, this model showed a reduction in the stimulated DA release upon amperometry analysis, suggesting altered dopaminergic neurotransmission. The electrophysiological analysis of the glutamatergic transmission showed that LTP induction was normal but LTD was absent. Furthermore, the mice exhibited defects in their locomotor activity in the open field, rotarod and startle tests at 3 months of age [99]. Later, in 2007, Manning-Boğ and colleagues generated another *Dj-1* KO mouse model that targeted the intron between exons 6 and 7. They reported that these *Dj-1* KO mice showed no change in the number of TH-positive and Nissl-stained nigral cells compared to the WT mice, and no change in the levels of DA and its metabolites, DOPAC or homovanillic acid (HVA). Moreover, no behavioral impairment was observed. However, the mice showed alterations in their striatal DA transmission: a marked increase in striatal DAT was detected in synaptosomal fractions from null *Dj-1* mice, and this increase in cell surface DAT was confirmed by autoradiographic studies [108].

Altogether, studies on *Dj-1* KO mice converge in concluding that *Dj-1* KO mice do not show the hallmark of PD found in *DJ-1* mutated patients, i.e., the loss of SNc DA neurons and the loss of nigrostriatal fibers. The underlying reason could be genetic compensation, a frequent phenomenon in the KO mouse model [48]. However, because subtle dysfunction features were observed, these models might be useful to study early dysfunction induced by the loss of DJ-1 function.

### 4.2. Rat Model

Under the sponsorship of The Michael J. Fox Foundation for Parkinson’s Research, which supports the generation, characterization and distribution of rat genetic models to accelerate PD research [122], *Dj-1* KO rats were generated using ZFN Technology [34,74]. The *Dj-1* KO rats showed a 50% loss of SNc DA neurons at 8 months of age [34,109]. The *Dj-1* KO rats also showed a marked reduction in TH-positive neurons in the LC at 8 months of age [110]. The TH immunoreactivity in the striatum was unchanged [34]. In the striatum, ultra-high performance spectroscopy showed an increase in the DA and 5-HT content at 8 months [34]. The quantitative radiography analysis of the striatum showed no change in the DAT density, but an increase in VMAT2, D_1_, D_2_, and D_3_ DA receptors between 4 and 8 months of age [74,109]. *Dj-1* KO rats also showed an increase in bodyweight [110,111]. Behavioral tests showed motor impairment in their movement, strength and coordination between 6 and 8 months of age [34]. The cylinder test revealed that *Dj-1* KO rats took more forelimb and hindlimb steps than their WT counterparts starting from 4 to 13 months of age, whereas the adhesive removal test disclosed no differences in their sensorimotor function [111]. *Dj-1* KO rats showed no anxiety or depression at 4, 8 or 17 months of age when tested with an elevated-plus maze or with the sucrose preference test [111]. A decrease in sucrose preference is an indication of anhedonia, which is a core feature of depression. The *Dj-1* KO rats showed no signs of anhedonia, and indeed they drank more sucrose with respect to the WT, with and without stress playing a factor, which probably indicates some abnormality in the neuroendocrine system [111]. In conclusion, while *Dj-1* KO mice do not show nigral degeneration [99,103,106], *Dj-1* KO rats exhibit progressive nigral neurodegeneration and motor impairments. This difference is in line with findings from rat models based on the genetic modification of α-syn. The overexpression of mutated *Snca* induces severe DA loss in the rat, whereas a similar effect has not been reported after the overexpression of the same gene in mice [123]. The underlying reasons are unclear; a plausible explanation is that these differences stem from the physiological, anatomical, biochemical and pharmacological differences between rats and mice [124].

### 4.3. Zebrafish (Danio rerio)

Zebrafish *dj-1* is evolutionarily conserved: it is 83% identical and 89% similar to human *DJ-1*, and is widely expressed in neurons [125]. In adult zebrafish brains, Dj-1 is present in the DA neurons of the olfactory bulbs, the telencephalon and each of the DA nuclei of the diencephalic groups [125]. In addition, Dj-1 is expressed in the LC and other hindbrain catecholaminergic neuronal groups [125]. Bretaud and colleagues, in 2007, generated KO zebrafish *dj-1* using the microinjection of antisense MO oligonucleotides into zebrafish embryos [112]. The *dj-1* KO zebrafish embryos showed no loss of DA neurons in the basal condition, but a pronounced reduction in the number of DA neurons was evident after exposure to H_2_O_2_ or proteasome inhibitor MG132. *Dj-1* KO had increased levels of p53 and the proapoptotic protein Bax, and MG132 exposure further enhanced the p53 and Bax expression in *dj-1* KO zebrafish embryos [112]. Later, in 2009, Baulac and colleagues generated another *dj-1* KO zebrafish model with MO oligonucleotides directed against the 5′ untranslated region of *dj-1*. This model was similar to Bretaud’s model and it showed no change in the DA neurons of embryos, but it did show greater sensitivity to exposure to H_2_O_2_, with an increase in the number of apoptotic cells, as observed by TUNEL (terminal deoxynucleotidyl transferase dUTP nick end labeling) staining [113].

In 2019, Edson and colleagues used the CRISPR-Cas9 method to generate a *dj-1* KO zebrafish model. This *dj-1* KO zebrafish model underwent normal development until the adult stages, was viable and fertile, but beyond age 3 months the fishes tended toward a smaller body size and lower mass [114]. With age (16 months), there was a severe reduction in the TH levels and DA content in the brain, and in the complex I activity in the skeletal muscle. A proteomic analysis of the early adult brains revealed that about 5% of the 4091 identified proteins were influenced by the lack of Dj-1. The dysregulated proteins were mainly those involved in mitochondrial metabolism, mitophagy, stress response, redox regulation and inflammation [114].

Hughes and colleagues, in 2020, used an unbiased computational method to record the movement of *dj-1* KO fishes and found that the *dj-1* KO zebrafish displays an overall reduction in movement resembling bradykinesia: a reduction in the distance travelled, velocity, time spent moving and duration of a swimming episode. In order to investigate the changes in gene expression associated with Dj-1 loss, an RNA-seq on adult zebrafish brains disclosed many metabolic changes [115]. Overall, these results suggest that the zebrafish is a useful model for studying the functions of Dj-1 in vivo and the consequences of *dj-1* mutations.

### 4.4. Drosophila melanogaster

*Drosophila melanogaster* has two genes—*dj-1α* (CG6646) and *dj-1β* (CG1349)—that share substantial homology with human *DJ-1* (*dj-1α* shares 56% identity and 70% similarity with the human protein; *dj-1β* shares 52% identity and 69% similarity) [116,119]. In adult flies, dj-1β is expressed at similar levels in male and female heads, brains and bodies, while dj-1α is expressed at detectable levels only in males, in the testes in particular, and at low levels in the adult head [116,119]. In order to address the biological function of dj-1 in PD with the *Drosophila* model, studies were conducted using *dj-1β* KO or double-KO (DKO) [116,117,119]. *Drosophila dj-1β* KO demonstrated the extended survival of DA neurons and resistance to oxidative stress induced by paraquat [116,117]; however, it showed increased sensitivity to H_2_O_2_-induced toxicity [116,117]. In addition, *dj-1β* KO displayed reduced taste sensitivity under normal food conditions with behavioral feeding assays, defects in the ability to form associative memories with taste memory assays [118], and decreased climbing ability [117]. Although *dj-1β* KO showed resistance to exposure to paraquat stress, a further loss of climbing activity occurred after repeated paraquat exposure [117]. The DKO lines were viable and fertile, and they displayed a normal lifespan. Though the model did not show differences in the number of DA neurons, it displayed increased sensitivity to chemical agents like H_2_O_2_ and paraquat, and much higher sensitivity to rotenone [119]. Hence, *dj-1* mutant flies can be considered an interesting model to study motor impairment in DJ1-associated PD.

### 4.5. Caenorhabditis elegans

*C. elegans* possesses two distinct *dj-1* paralogs: *djr-1.1* and *djr-1.2* [120]. Ved and colleagues, in 2005, generated a KD of the *djr-1.1* gene [45] and found that *C. elegans* KDs for the *djr-1.1* gene were far more sensitive to rotenone exposure than the control nematodes. Therapies that stimulated complex II function and inhibited apoptosis (D-β-hydroxybutyrate and tauroursodeoxycholic acid) provided protection against complex I inhibition [45]. Another group generated double mutants for the *djr-1.1* and *djr-1.2* genes. They exposed *C. elegans* to *Pseudomonas aeruginosa* and investigated their innate immunity. They found that the loss of *djr-1.1* and the *djr-1.2* function stimulated the phosphorylation of p38MAPK/PMK-1, a key step in inflammatory signaling. The findings underscored the importance of DJ-1 for the regulation of innate immune responses, and they suggested that DJ-1 loss-of-function may promote neuroinflammation in PD [120].

Another research group generated worms with *djr* deletions: *djr-1.1* or *djr-1.2* alone, and *djr* double deletion [121]. The researchers showed that the deletion of *djr-1.2* alone or *djr-1.1* and *djr-1.2* together led to a marked increase in sensitivity to Mn and decreased survival and lifespan after acute Mn exposure. *Djr-1.2* overexpression was sufficient to restore protection against Mn-induced lethality in *djr* double deletion mutants [121]. In addition, the overexpression of DAF-16, the *C. elegans* ortholog of human FOXO, which functions as a transcription factor for antioxidant response, fully reversed the lifespan shortening caused by *djr-1.2* deletion and Mn exposure [121]. In order to investigate the role of DJR-1.2 in DA signaling, the researchers employed an assay to detect spontaneous movement in the dauer state, and they observed that *djr-1.2* deletion alone greatly increased dauer movement; this phenomenon is compatible with reduced dopaminergic signaling. Furthermore, *djr-1.2* deletion combined with Mn exposure increased the dauer movement, suggesting that DA signaling is further impaired by Mn exposure [121].

## 5. *PARK9*: The ATPase 13A2 Gene (ATP13A2)

A mutation of the *ATP13A2* gene (OMIM 606693), located on the 1p36.13 locus and first described in 1994, was associated with a rare hereditary form of autosomal recessive juvenile onset Parkinsonism in the Kufor-Rakeb community in Jordan [126]. The *ATP13A2* gene encodes a transmembrane endo/lysosomal-associated protein belonging to the P5 type transport ATPase subfamily [127,128,129]. ATP13A2 is ubiquitously expressed, with the strongest expression in the brain and in the SNc [129]. Studies have shown that ATP13A2 is involved in autophagy and the lysosomal pathways [130,131]. It plays a role in the metabolism of Mn^2+^ and Zn^2+^ [127,132,133], and in mitochondrial pathways [134]. ATP13A2 can also regulate the metabolism of α-syn, one of the major components of Lewy bodies [135] (Table 4).

### 5.1. Mouse Model

*Atp13a2* KO mice were created to study the role of ATP13A2 in PD. *Atp13a2* KO mice grew normally and were fertile. The sensorimotor function in this model was measured using tests sensitive to various types of motor dysfunction, including challenging beam traversal, spontaneous activity, gait, and nest building. The sensorimotor function did not differ between the genotypes at 5–12 months of age; however, at 20–29 months the *Atp13a2* KO mice displayed severe impairment on all of the motor tests [136]. Histopathology showed no change in the number of DA neurons in the SNc or in the striatal DA levels in aged *Atp13a2* KO mice [136,137]. Nonetheless, the model disclosed many progressive neuropathological changes: the early onset of gliosis at 1 month of age, lipofuscinosis at 3 months, the accumulation of lysosome-associated membrane glycoprotein (LAMP) 1 and 2, and lysosomal lipid bis(monoacylglycero)phosphate (BMP) at 6 months [131,135,136,137]. At 12 months of age, the *Atp13a2* KO mice showed the aggregation of ubiquitinated proteins and p62 and aberrant processing of the lysosomal protease cathepsin D (CATD) in many brain regions [137]. The *Atp13a2* KO mice also displayed the accumulation of insoluble α-syn in the hippocampus at 18-20 months, but no change in the α-syn protein solubility in either the cortex or the cerebellum [136]. The *Atp13a2* KO mice did not present a substantial impairment of their spontaneous activity between 5 and 15 months of age, while differences emerged on the challenging beam traversal and the gait test between 20 and 29 months of age [136].

### 5.2. Zebrafish (Danio rerio)

The zebrafish has two proteins with high homology with human ATP13A: zgc:136762 and zgc:63781, with 1170 and 1177 amino acids, respectively; zgc:136762 is closely related to human ATP13A2, with 50% homology and 69% similarity, while zgc:63781 is phylogenetically closer to human ATP13A1, sharing 73% identity and 83% similarity. The human ATP13A2 has two functional domains, an E1–E2 ATPase domain and a haloacid dehalogenase (HAD-like) domain, which are important for the biological role of the protein. Both domains are present in the zebrafish zgc:136762 sequence. The Atp13a2 protein is widely expressed throughout the body of the embryo; during embryonic development, its expression becomes more restricted to the brain area [138]. In order to obtain zebrafish *atp13a2* KO, MOs were used to inhibit the correct splicing of *atp13a2* mRNA: the complete abrogation of Atp13a2 led to embryonic lethality, while the partial abrogation of atp13a2 allowed the offspring to survive. Partial gene KD embryos showed a postural defect, i.e., a curved phenotype at 48 h post-fertilization (hpf). In addition, the KD of *atp13a2* in embryos at 7 dpf caused a distinct movement pattern characterized by general movement latency and an abnormal response to environmental stimuli, as observed by the swimming pattern/locomotor activity in the light on–light off paradigm [138].

Heins-Marroquin and coauthors used zebrafish lines carrying *atp13a2* mutant alleles (*atp13a2^sa14250^* or *atp13a2^sa18624^*) generated by ENU mutagenesis. Each allele carries a single non-sense mutation (2153 T > A or 2457 T > G) resulting in a premature stop codon in exon 20 or exon 22, respectively. Both mutations were predicted to cause loss-of-function. Homozygous *atp13a2^sa18624^* mutants were able to reach adulthood without any obvious morphological or behavioral abnormalities [139]. As Atp13a2 seems to play an important role in heavy metal homeostasis, the authors tested the effect of Mn on *atp13a2^sa18624^*^+/+^, *atp13a2^sa18624−/+^* and *atp13a2^sa18624−^*^/*−*^ larvae. Exposure to MnCl_2_ was found to cause neurodegeneration in zebrafish [144]. At 5 dpf, in the presence of Mn^2+^, WT and heterozygous larvae showed a moderate phenotype characterized mainly by an underdeveloped swimming bladder. In contrast, the homozygous *atp13a2* mutants were highly sensitive to Mn^2+^ and displayed multiple abnormalities in addition to swimming bladder underdevelopment, including pericardial edema, movement loss and spinal curvature. Similar results were obtained in the second mutant line (*atp13a2^sa14250^*) upon exposure to Mn^2+^. Furthermore, compared to WT, 64% of the *atp13a2^sa18624^*^−/−^ larvae showed large apoptotic areas throughout the CNS (brain and dorsal spine) after exposure to Mn^2+^ [139].

Recently, another *atp13a2* deficient zebrafish was generated by introducing a 10 bp deletion in exon 2 of *atp13a2* using CRISPR/Cas9 gene editing [140]. This *atp13a2* KO zebrafish model showed, at 4 and 12 months of age, a severe loss of DA neurons in the posterior tuberculum (64% at 4 months and 37% at 12 months) and norepinephrine neurons in the LC (52% at 4 months and 40% at 12 months), as observed by the immunostaining of TH-positive neurons [140]. Furthermore, the Western blotting, electron microscopy and liquid-chromatography tandem mass spectrometry (LC-MS/MS) of the brain tissues revealed that the *atp13a2* KO zebrafish had CATD deficiency, lysosomal dysfunction and intracellular vesicle trafficking dysfunction, which are all relevant to PD [140].

### 5.3. Medaka Fish (Oryzias latipes) 

The medaka fish has also been used to study Atp13a2. By screening the TILLING library, Matsui and colleagues found a mutation, “IVS13, T-C, +2”, [141]. The mutation resulted in an abnormal splicing pattern, which was almost identical to that observed in human *PARK9* patients in which 111-bp exon 13 has been skipped, causing the delocalization of the Atp13a2 protein in the endoplasmic reticulum rather than in the lysosome [10,141]. Atp13a2 “IVS13, T-C, +2” mutant medaka had a shorter lifespan and exhibited more spontaneous swimming movement at 4 months but no differences in swimming at 12 months. They showed a reduction in the number of TH-positive neurons in the middle diencephalon and in the TH-positive fiber density in the telencephalon at 8 and 12 months. Furthermore, a reduction in their DA content was observed at 12 months. At 8 and 12 months, there were fewer noradrenergic neurons in the medulla oblongata [141]. Mutant medaka disclosed decreased CATD protein levels and activity, and fingerprint-like subcellular structures in the brain, indicating an autophagy/lysosome disorder [141].

### 5.4. Caenorhabditis elegans

The ortholog of the human *ATP13A2* gene in *C. elegans* is the *W08D2.5*/*catp-6* gene. The *catp-6*(*ok3473*) mutant bears a deletion of 900 bp that results in a substantial loss of *catp-6* mRNA [142]. *C. elegans catp-6* mutants showed decreased locomotion, a severe delay in the rate of development [142] and higher mortality in midlife [143]. Furthermore, *catp-6* mutants displayed an alteration in their iron homeostasis caused by the major down-regulation of the core genes required for metabolizing iron [142]. Young (L4 larvae to day 2 adults) and middle-aged (day 4 to day 5 adults) *catp-6* mutants also displayed altered Zn homeostasis [143].

In order to assess the autophagy and lysosomal function in these mutants, Anand and coauthors quantified the protein levels of autophagosome-(cleaved LGG-1/LC3-II) and lysosome-specific aspartyl protease ASP-3/CATD, respectively. The protein level of cleaved LGG-1/LC3-II was severely reduced in the *catp-6* mutants, suggesting a defect in their early autophagosome formation. The lysosomal function was assessed by monitoring the levels of the premature (P-CATD) and the mature (M-CATD) form of the pH-dependent lysosomal enzyme aspartyl protease ASP-3/CATD: the P-CATD levels were far higher in the *catp-6* mutants than in the WT worms, suggesting the reduced conversion of P-CATD to M-CATD. The *Catp-6* mutant worms also showed the low mRNA expression of several key genes required for autophagy and lysosomal function [142]. Altogether, these results show that the loss of *catp-6* affected autophagy and lysosomal function. Furthermore, the *catp-6* mutants presented multiple defects in their mitochondrial function, including a reduced mitochondrial membrane potential and maximal respiration rate, as well as increased sensitivity to rotenone, which could be rescued by treatment with iron chelators [142].

## 6. *PARK14*: The Phospholipase A2 Group VI Gene (*PLA2G6*)

The phospholipase A2 group VI (*PLA2G6*) gene (OMIM 612953) is located on chromosome 22 (22q13.1); it encodes a calcium-independent phospholipase A2 enzyme (PLA2G6) which is widely expressed in the brain, particularly in dendrites and axon terminals [145]. Mainly involved in the homeostatic processes of cell membranes, the protein selectively hydrolyzes glycerophospholipids to release free fatty acids [146]. Recessive mutations in *PLA2G6* were found to induce alterations in phospholipid metabolism and abnormal iron accumulation, which ultimately led to infantile neuroaxonal dystrophy (INAD), atypical neuroaxonal dystrophy (ANAD) and neurodegeneration with brain iron accumulation (NBIA). These diseases cause neurodegeneration in early life [147]. *PLA2G6* mutations (e.g., D331Y, R365Q, R714Q, R747W) were also found to be associated with familial forms of PD (*PARK14*) characterized by AR inheritance and early-onset dystonia-Parkinsonism [148]. Among these mutations, the compound heterozygous mutations p.F72L/p.R635Q and p.Q452X/p.R635, and the homozygous mutation D331Y are associated with AR EOPD [149,150]. Other less common mutations associated with *PARK14* phenotypes are R37X, F72L, A80T, M358I, Q452X, A499T, T572I, R632W, N659S, D739H and P806R [151]. Generally, *PARK14* patients show psychomotor deterioration, cerebellar ataxia and dystonia, as well as tremors, rigidity, bradykinesia and autonomic dysfunction [151]. Immunohistochemical studies have shown the presence of Lewy bodies, neurofibrillary tangles and neuropil threads, and the loss of DA neurons in the SNc, along with the accumulation of phosphorylated α-syn in the nerve cell bodies and processes of the gastrointestinal tract and cardiac sympathetic nerves [151] (Table 5).

### 6.1. Mouse Model

In 2008, Shinzawa and colleagues generated and characterized *Pla2g6* KO mice [152]. In their first year of life, the animals showed no developmental deficits, but their fertility was reduced (especially among males). By the age of 2 years, motor dysfunctions (especially the hindlimb clasping reflex during tail suspension), abnormal gait and poor performance in the hanging wire grip test were observed [152]. The *Pla2g6* KO animals showed the degeneration of nigrostriatal DA neurons and the focal loss of TH and DAT nerve terminals in the striatum (from 56 weeks of age) [152]. The *Pla2g6* KO animals also showed prominent axonal degeneration and the widespread presence of spheroids and vacuoles throughout the CNS and the peripheral nervous system (PNS) [153]. In *Pla2g6* KO mice, Sumi-Akamaru and colleagues observed atrophic axons accompanied by mild neuronal loss in the late stages of the disease [154]. They reported the strong expression of α-syn in the mitochondrial outer membrane protein (TOM20)-positive neuronal granules before the onset of motor dysfunction [154].

In 2010, Basselin and coworkers investigated whether the brain metabolism of docosahexaenoic acid (DHA), the substrate of the PLA2G6 enzyme, was reduced in the early stages of the disease. To this aim, 4-month-old *Pla2g6*^−/−^, *Pla2g6*^+/−^ and *Pla2g6*^+/+^ mice were administered radiolabeled unesterified [1-14C] DHA, and its metabolism (incorporation coefficients and rates) was determined by quantitative autoradiography. *Pla2g6*^−/−^ and *Pla2g6*^+/−^ mice showed a severe baseline reduction in DHA metabolism compared to the *Pla2g6*^+/+^ mice, including 4-month-old mice which lacked histopathological or neurological alterations [155].

Zhou and colleagues developed a novel transgenic mouse model with the constitutive deletion of exon 2 of the *Pla2g6* gene (*Pla2g6* ex2^KO^) resulting in the expression of a truncated ex2^KO^ Pla2g6 protein which lacks the first 178 amino acids in the N-terminus [156]. The genetic truncation of the N-terminus did not affect the protein catalytic activity [156]. Mouse embryonic fibroblasts (MEFs) from *Pla2g6* ex2^KO^ mice exhibited functional impairment in stored–operated Pla2g6-dependent Ca^2+^ signaling, with the substantial depletion of intracellular Ca^2+^ stores. *Pla2g6* ex2^KO^ mice developed progressive age-dependent motor dysfunction starting from 16 months of age. The mice displayed impairment in motor coordination, e.g., a progressive age-dependent increase in the number of missteps on the balance beam test; the results from the pole and the rotarod test further confirmed severe PD-like motor dysfunction in these animals. Interestingly, the *Pla2g6* ex2^KO^ mice were responsive to L-DOPA: the administration of L-DOPA markedly improved their motor coordination in an age- and dose-dependent manner. The analysis of the SNc area of the brain from aged *Pla2g6* ex2^KO^ animals revealed a progressive age-dependent loss of SNc DA neurons: no changes were observed at 8 months of age, whereas over 30% of the DA neurons in the SNc of ex2^KO^ mice was lost by 16 months, and over 50% by 24 months of age. In addition, the DA neurons in the SNc of PLA2g6 ex2^KO^ showed marked autophagy dysfunction, i.e., the accumulation of LC3 and an increased autophagosome number [156].

In 2019, Chiu and colleagues generated a KI mouse model harboring the homozygous human *PLA2G6* D331Y (GAC→TAC) mutation, which associates with Parkinsonism in humans (*Pla2g6^D331Y^*^/*D331Y*^). The aim was to study the neurological and neuropathological changes associated with this mutation, as well as the molecular mechanisms leading to DA neuron degeneration [157]. The model showed early-onset neuronal death in the SNc, as observed by a reduction in the TH-positive neurons in the SNc at 6 and 9 months of age and a decreased density of striatal TH staining at 9 months, indicating the degeneration of nigrostriatal DA terminals. Lewy bodies were found in the SNc of 9-month-old *Pla2g6^D331Y^*^/*D331Y*^ mice, along with the upregulated protein expression of α-syn and phosphorilated α-syn, while 6- to 12-month-old *Pla2g6^D331Y^*^/*D331Y*^ mice exhibited a progressive PD phenotype characterized by slowness of movement, hypoactivity, and impaired motor coordination and performance. L-DOPA treatment reversed the hypoactivity. A homozygous D331Y *Pla2g6* mutation led to abnormalities in the mitochondrial ultrastructure in the SNc, i.e., the disrupted structure of the mitochondrial cristae and smaller mitochondria in the *Pla2g6^D331Y^*^/*D331Y*^ mice compared to the WT mice. The model revealed multiple mitochondrial dysfunctions in the SNc, including decreased mitochondrial complex I and III activity, a lower ATP level, increased ROS production and lipid peroxidation, and the upregulation of the cytosolic level of cytochrome c. In addition, homozygous D331Y *Pla2g6* mutation activated the mitochondrial apoptotic pathway and induced endoplasmic reticular stress and mitophagy dysfunction. The study demonstrated that homozygous D331Y *Pla2g6* mutation was able to cause DA neuron degeneration in the SNc and induce an early-onset PD phenotype in the mice [157].

### 6.2. Drosophila melanogaster

*Drosophila melanogaster* was used to study the pathological effects of *pla2g6* mutations. iPLA2-VIA, the *Drosophila* homolog of *PLA2G6*, shows 51% identity and 67% positivity with the human gene [159]. Lin and colleagues generated a KO fly model of *pla2g6* (iPLA2-VIA KO). While the lack of iPLA2-VIA did not modify the phospholipid composition of the brain, it increased the production of ceramides (thus affecting membrane fluidity), which led to impaired synaptic transmission and neurodegeneration, resulting in a shortened lifespan [146]. In 2019, Mori and colleagues reported impaired neurotransmission in the early developmental stages of iPLA2-VIA KO flies, which was associated with the progressive degeneration of DA neurons and the formation of α-syn aggregates [158].

## 7. *PARK15*: The F-Box Protein 7 Gene (*FBXO7*)

Homozygous or compound heterozygous mutations in the *F-box only protein 7* (*FBXO7*) gene located on chromosome 22q12.3 are associated with Parkinsonian-pyramidal syndrome (PPS) and AR EOPD (*PARK15*, OMIM 260300) [160,161]. *FBXO7*-mutated patients show early-onset bradykinesia, tremors, rigidity, pyramidal tract signs and L-DOPA response [160,161,162,163]. *FBXO7* encodes F-box protein 7 (FBXO7), which is a member of the F-box-containing protein (FBP) family. These proteins function as adaptors for the SKP1/cullin-1/F-box protein E3 ubiquitin ligase complex to recognize substrates and facilitate their ubiquitination [160]. FBXO7 is broadly expressed in human tissues, but is particularly enriched in the bone marrow, liver, kidney, testis and thyroid gland. FBXO7 is also expressed in several regions of the human brain: its expression is high in the cerebral cortex, globus pallidum, thalamus, striatum and substantia nigra, but low in the hippocampus and cerebellar cortex [164]. PD-associated mutations in *FBXO7* can induce impairment in mitochondrial homeostasis by disrupting mitochondrial mitophagy. FBXO7 deficiency is also associated with an increased mitochondrial NADH redox index, the impaired activity of complex I in the ETC, reduced mitochondrial membrane potential and ATP contents, and increased cytosolic ROS production [165,166] (Table 6).

### 7.1. Mouse Model

Vingill et al. generated a conventional *FbxO7* KO mouse in which the fourth exon of the *FbxO7* gene was deleted, resulting in a truncated protein [167]. At postnatal day (P) 5, there was no difference in the body or brain weight between the *FbxO7* KO and the control littermates, while at P18 the *FbxO7* KO mice displayed a substantially lower body and brain weight, and obvious motor deficits. The *FbxO7* KO mice died a premature death starting from postnatal week 4 [167]. At P18, the *FbxO7* KO mice displayed a moderate increase in cell death in the cortex, but no difference in TH-positive neurons in the SNc or change in the DA level and its metabolites in the striatum. Further neuropathological investigation of the *FbxO7* KO brain revealed no a-syn protein deposits, though astrogliosis was observed in the cortex, because the area occupied by the glia fibrillary acidic protein (GFAP)-positive astrocytes was increased [167].

Because the systemic loss of *FbxO7* caused premature death, perhaps due to a negative impact on the peripheral organs, the authors generated and characterized conditional *FbxO7* KO mice. In detail, they first generated *FbxO7*-floxed animals (*FbxO7^fl^*^/*fl*^) carrying loxP-flanked exon 4, then homozygous conditional *FbxO7^fl^*^/*fl*^ animals were bred to a NEX (neuronal helix-loop-helix protein-1)-Cre line [169] to delete *Fbx07* from the pyramidal neurons of the cortex and the hippocampus (Nex-Cre;fl/fl mice) or B6.Cg-Tg(TH-cre) 1Tmd/J line [170] to delete the *Fbx07* gene from the TH-expressing neurons (TH-Cre;fl/fl mice) [167]. The conditional Nex-Cre;fl/fl mice were viable and lived for at least 4 months. They showed a normal bodyweight at 2 months of age, after which their growth stagnated. At 2 and 4 months, they displayed spasticity and progressive motor coordination deficits, which resembled the pyramidal tract signs seen in *PARK15*-mutated patients. The mice also showed an increase in GFAP-positive cells and astrogliosis in the cortex [167]. TH-Cre;fl/fl mice lived longer than the previous mice, because most of them lived for at least 12 months. At age 2 months, the mice showed no motor deficit, but later (at 6 months) motor deficits began to appear on the rotarod test; by 12 months their overall health and motor performance had further worsened. The TH-Cre;fl/fl mice displayed slow movements, reduced mobility and alterations in many fine gait parameters; however, unlike the Nex-Cre;fl/fl mice, they retained their coordination ability. The deletion of Fbxo7 in the catecholaminergic neurons caused symptoms reminiscent of the bradykinesia and rigidity seen in *PARK15* patients. The histological analysis of the brain revealed no changes in the number of DA neurons in either the 2-month-old or the 12-month-old TH-Cre;fl/fl mice, though there was a 50% reduction in the DA content in the striatum at 2 and 12 months of age. Neuropathological examination revealed an increase in astrogliosis in the SNc [167].

In conclusion, although the systemic and specific neuronal loss of the *Fbxo7* gene in mice does not cause DA neuron degeneration, severe motor deficits reminiscent of the symptoms seen in *PARK15* patients were observed in the conventional and the conditional *FbxO7* KO mice. These models are therefore useful to study early- and late-onset motor impairments characterizing the PPS associated with mutations in the *FBXO7* gene.

### 7.2. Zebrafish (Danio rerio)

In the zebrafish genome, a single homolog to human *FBXO7*, termed *zFbxo7*, is annotated. The zFbxo7 protein is expressed throughout embryonic development until adulthood; it is abundantly present in the brain and the liver, but scarce in the heart and the kidney. It is ubiquitously expressed in the brain: high levels are found in the neurons of the olfactory bulb and the diencephalon, intermediate levels in the cerebellum and the medulla oblongata, and low levels in the optic tectum and the habenula [168]. The first vertebrate animal model of *PARK15* was generated by Zhao and colleagues in 2012 by the MO-mediated KD of *zFbxo7*; it is the only zebrafish model created to date [168]. These researchers generated two non-overlapping *zFbxo7* MOs, one targeting the ATG translation initiation site (ATG-MO) and the other targeting the exon2/intron2 splice site (SP-MO) of *zFbxo7*, respectively. Both MOs were able to knock down zFbxo7 protein expression. A range of morphological phenotypes, spanning from mild (ATG-MO-Mild and SP-MO-Mild morphants) to severe (ATG-MO-Severe and SP-MO-Severe morphants), was observed at 72 hpf, including curly tails, heart edema and heart malformations; the severity of the phenotype was correlated with the silencing of the zFbxo7 protein. A significant reduction (40%) in the number of *TH-positive* neurons was seen in the SP-MO-severe morphants compared to the WT zebrafish, while the number of *DAT-positive* neurons was considerably reduced in the ATG-MO and the SP-MO morphants, and more dramatically in the morphants with a more severe phenotype (ATG-MO-Severe and SP-MO-Severe) [168]. Because the ATG-MO-Severe and SP-MO-Severe morphants showed scarcely any motor activity, the researchers focused the locomotor analysis on the ATG-MO-Mild and SP-MO-Mild phenotypes: the morphants displayed a markedly decreased swimming velocity, which was improved by exposure to the DA agonist apomorphine [168]. Because it reproduces the pathological (DA neuronal loss) and behavioral (locomotor disturbances) hallmarks of human Parkinsonism, the model provides a valid tool for investigating the mechanisms underlying selective DA neuronal death in AR EOPD.

### 7.3. Drosophila melanogaster

Zhou and colleagues generated a transgenic *Drosophila* model overexpressing human WT *FBXO7* in the DA neurons [166]. The model had a normal lifespan, but by 30–40 days began to show locomotor deficits in its flight and climbing ability. At 40 days there was a loss of DA neurons, especially in the paired posterior medials 1 and 2 (PPM1/2) and paired posterior medial 3 (PPM3) DA neuron clusters [166]. The overexpression of the human WT FBXO7 protein also led to fbxo7 protein aggregation and a mitochondrial deficit, with the swelling of the muscle mitochondria, broken mitochondrial cristae, and the accumulation of high-density materials in the swollen mitochondria [166]. Thus, the results suggest that the overexpression of FBXO7 can be detrimental to DA neurons in *Drosophila*.

## 8. *PARK19*: The DnaJ Heat Shock Protein Family (Hsp40) Member C6 Gene (*DNAJC6*)

The *DNAJC6* (DnaJ (Hsp40) homolog, subfamily C, member 6) gene (OMIM 615528) is located on chromosome 1p31.3 and encodes the brain-specific isoform of AUXILIN. AUXILINS have a well-established role in clathrin-mediated endocytosis (CME), which is responsible for the uptake of material into the cells through clathrin-coated vesicles [4,171]. Mutations of *DNAJC6* are associated with AR EOPD [171,172,173]. Homozygous *DNAJC6* c.801-2A > G mutation was discovered in 2012 in two patients of a consanguineous family with ARJP [172]. Other disease-associated *DNAJC6* variants were discovered, including p.Q734X, p.R927G, c.2223A > T, p.L209P, p.M133L, p.R619C, p.F839Lfs*22 and c.2038 + 3A > G [171,173,174]. The AUXILIN-1 protein has a high homology with the cyclin G-associated kinase (GAK) protein. Though they share a multidomain structure, GAK has an additional N-terminal kinase domain, and it is ubiquitously expressed, whereas AUXILIN-1 is neuron-specific and enriched in the nerve terminals [175,176,177]. Both AUXILIN-1 and GAK are members of the J domain family of proteins, which are characterized by the canonical histidine-proline-aspartic acid motif that binds to HSC70 [175,176,177] (Table 7).

### 8.1. Mouse Model

Conventional *Dnajc6* KO mice were generated in 2010 [177]. The rate of early postnatal mortality among the mutant mice was high, but the surviving pups had an apparently normal lifespan. The birth weight of the KO and heterozygous mice was lower than that of WT pups. Female *Dnajc6* KO mice also showed delayed sexual maturity [177]. Because AUXILIN-1 and GAK are highly homologous proteins, Yim and colleagues studied the *Dnajc6* KO mice to determine whether GAK was up-regulated and could compensate for the absence of AUXILIN-1. A Western blot analysis showed that *Dnajc6* KO embryos and 3–5-week-old mice had higher GAK levels in the brain than WT mice. In contrast, the GAK levels in nonneuronal tissues (e.g., spleen, liver, kidney, testes) were not very different between the *Dnajc6* KO and WT mice [177]. The study also showed a relationship between the brain GAK levels and bodyweight: the bodyweight was lower in the *Dnajc6* KO pups with no up-regulation of GAK, whereas the *Dnajc6* KO pups with the up-regulation of GAK showed no change in their bodyweight compared to the WT [177]. However, the analysis of the brain tissue of 1-week-old *Dnajc6* KO mice showed no change in the endocytic synaptic proteins dynamin 1, synaptojanin 1, epsin, eps15, amphiphysin 1 or Hsc70 [177]. Because the study was performed before the identification of *DNAJC6* as the gene associated with AR EOPD, it did not investigate whether the *Dnajc6* KO mice displayed a loss of DA neurons or motor impairment.

### 8.2. Drosophila melanogaster

Two different RNAi lines in *Drosophila* were generated by silencing the *GAK* homolog *auxilin* (*aux*): *aux*R^16182^ and *aux*R^103426^ [178]. The downregulation of auxilin in *Drosophila* decreased the lifespan and caused an age-dependent reduction in the DA neuron number [178]. *Drosophila aux*R^16182^ and *aux*R^103426^ showed no obvious alteration in the mitochondrial morphology in the DA neurons, but did show increased sensitivity to oxidative stress upon paraquat exposure [178]. Furthermore, *aux*R^16182^ and *aux*R^103426^ displayed locomotor deficits in climbing ability [178]. The model therefore provides an interesting tool to study the consequences of *auxilin*/*GAK* silencing.

## 9. *PARK20*: The Synaptojanin 1 Gene (*SYNJ1*)

Mutations in the gene *SYNJ1* (OMIM 615530) are associated with AR EOPD [4,179,180,181,182,183]. Patients with a mutation in *SYNJ1* develop progressive tremors, dystonia, rigidity and cognitive decline [180]. The *SYNJ1* gene is located on 21q22.2 and encodes the protein SYNAPTOJANIN-1, an important phosphoinositide phosphatase enriched in nerve terminals that regulates synaptic vesicle recycling [180,184] (Table 8).

### Mouse Model

*Synj1* KO mice were generated in 1999. The KO mice were indistinguishable from WT mice at birth, but within a few hours they could be easily identified by the severely low milk content in their stomach: 85% died within 24 h and the remaining 15% died within 15 days. At 10 days of age, the *Synj1* KO mice weighed less than half of the weight the WT mice, showed severe weakness and ataxia, and displayed generalized convulsions upon stimulation by nociceptive stimuli [185]. The *Synj1* KO mice did not show abnormal levels of SYNAPTOJANIN-1 interactors (amphiphysin 1 and 2, Grb2, SH3P4, SH3P8, SH3P13), proteins involved in synaptic vesicle endocytosis (dynamin 1, clathrin, AP-2, AP180, Eps15, epsin1, auxilin, Hsc70), proteins implicated in the synaptic vesicle cycle (synapsin 1, rab3a, rab5, actin), and enzymes involved in phosphatidylinositol (PI) metabolism, i.e., PI(4)P 5-kinase type II and PI 3-kinase. Electron microscopy revealed structural changes in the synapses and many clathrin-coated vesicles in the cytomatrix-rich area surrounding the synaptic vesicle cluster. Electrophysiological experiments on the CA1 hippocampal area showed impaired synaptic vesicle recycling, with the depression of synaptic transmission after prolonged high-frequency stimulation and a corresponding delayed recovery after the interruption of the stimulus [185].

More recently, heterozygous *Synj1*^+/−^ mice were investigated as a potential model of early-onset PD. The *Synj1*^+/−^ mice had a normal bodyweight and lifespan [186]. At 7 and 12 months, they showed a reduction in DA, DOPAC, and HVA content in the striatum, as determined by HPLC analysis. Immunohistochemical analysis revealed a progressive reduction in DA terminals from 3 to 18 months of age, which reached 75% at 18 months [186]. Furthermore, immunofluorescence for phospho-Ser129 (pS129) α-syn showed an accumulation of pS129 α-syn in the cortex, the striatum and the midbrain in *Synj1*^+/−^ mice at 18 months [186]. *Synj1*^+/−^ mice also showed age-dependent motor impairment [186]. These findings suggest that *Synj1* heterozygous deletion leads to PD-like pathologies in mice.

## 10. *PARK23*: The Vacuolar Protein-Sorting 13 Homolog C Gene (*VPS13C*)

Vacuolar protein sorting (VPS) is evolutionarily conserved, and was identified in *Saccharomyces cerevisiae*. Human VPS13 (hVPS13) consists of a group of four ubiquitously expressed proteins (hVPS13A, hVPS13B, hVPS13C, hVPS13D). Although the molecular function of hVPS13 proteins is not yet well understood, studies on fibroblasts, neurons differentiated from patients’ fibroblasts, and organisms such as yeast and fruit flies all suggest that they might play a role in a variety of processes, including the regulation of vesicle trafficking, cell-to-cell interactions, mitochondrial functions, actin cytoskeleton organization and autophagy [187,188,189].

Several mutations identified in *hVPS13* genes have been associated with the onset of neurological and developmental disorders. Mutations in the *hVPS13A* (9q21.2) and *hVPS13B* (8q22.2) genes lead to recessive autosomal genetic disorders, including chorea-acanthocytosis (ChAc) and Cohen’s syndrome, respectively [190,191,192]. Mutations in *hVPS13D* (15q22.2) are associated with ataxia/spastic paraplegia. Polymorphisms or mutations of the *hVPS13C* (1p36) gene have been found to be associated with an increased risk of type 2 diabetes or young adult-onset PD (*PARK23*, OMIM 616840), respectively [193].

Notably, the link between *hVPS13C* mutations and *PARK23* is a recent discovery. A genome-wide association study (GWAS), involving a cohort of 13,708 PD patients and 95,282 controls, identified the *hVPS13C* gene as a new risk variant for PD in Caucasian and East Asian populations [194]. Lesage and colleagues performed homozygosity mapping and exome sequencing studies of 66 individuals with PD (23 family members and 43 isolated subjects) and 39 unaffected relatives, and discovered a homozygous T-to-G transversion in the *hVPS13C* gene (intron 61) of a Turkish woman with AR EOPD, and heterozygous truncating mutations (4-bp insertion in exon 11 and G-to-T transversion in exon 69 or 1-bp deletion in exon 43 and G-to-C transversion in exon 37) in the *hVPS13C* gene of two French individuals with EOPD. The individuals harboring the *hVPS13C* mutations showed a peculiar form of EOPD characterized by rapid, severe disease progression and early cognitive decline. The post-mortem examination of the brain showed numerous Lewy bodies in the brainstem, limbic system and many cortical areas, typically observed in the brain of patients with α-synucleinopathy [193]. One year later, the whole-exome sequencing (WES) of 80 individuals with sporadic EOPD showed a heterozygous *hVPS13C* mutation in one of the patients [195]. Additional studies led to the discovery of a large homozygous genomic deletion (50 coding exons) in the *hVPS13C* gene in a patient with clinical symptoms of Parkinsonism [196]. Finally, a recent paper described seven new *hVPS13C* mutations in a large cohort of Chinese patients with EOPD [197]. Unfortunately, no animal model has been developed yet for VPS13C.

## 11. Discussion

From the experimental data presented in this review, it is clear that, to date, none of the developed experimental models can be considered a perfect model of PD. However, many of these models reproduce some of the key features of the disease and, for this reason, they can be considered as useful tools for the study of pathogenetic mechanisms and for the screening of neuroprotective therapies (Figure 2). In general, the murine KO models, which in theory—from the genetic point of view—are those that best reproduce AR transmission pathologies, are those that, unexpectedly, have often shown milder or even absent phenotypes. One plausible explanation is that, often, genetic compensation occurs in response to gene knocking-out [48]. Mouse models expressing genes bearing mutations associated with the disease have shown more evident PD-associated phenotypes, and for this reason they can be considered valid models to study the toxicity induced by mutant proteins. 

Studies in *Drosophila* mutant flies have often revealed that the deletion/silencing of genes associated with AR PD leads to a phenotype that mirrors many of the neuropathological features found in human tissues; in particular, in flies, the mitochondrial dysfunction phenotype is often evident. In this context, the data on PD models are in agreement with studies carried out on other fly models of neurodegenerative diseases that identify *Drosophila* as a useful model genetic system for the understanding of neurodegeneration [198].

Studies on the teleost fishes zebrafish and medaka have highlighted that, in these models, the deletion/silencing of genes associated with AR PD often leads to the spontaneous death of DA neurons or to a higher susceptibility to toxic substances. This key neuropathological hallmark of PD is often lacking in other organisms, raising the possibility that teleost fishes have a vulnerability of DA neurons more similar to humans.

Finally, *C. elegans*, the less-used model to date, has produced interesting results for PD modeling, although more detailed studies will be necessary to confirm the validity of these models.

In conclusion, although none of the models described mirror all the characteristics of PD, the efforts made in the last 20 years by many laboratories have allowed us to create and characterize a multitude of models, and each of these can be useful in selected contexts. Taking into account the strengths and limitations of each model, the use of multiple models remains the best strategy to validate a pathogenetic mechanism or to test a therapeutic approach.

## Figures and Tables

**Figure 1 biomedicines-09-00812-f001:**
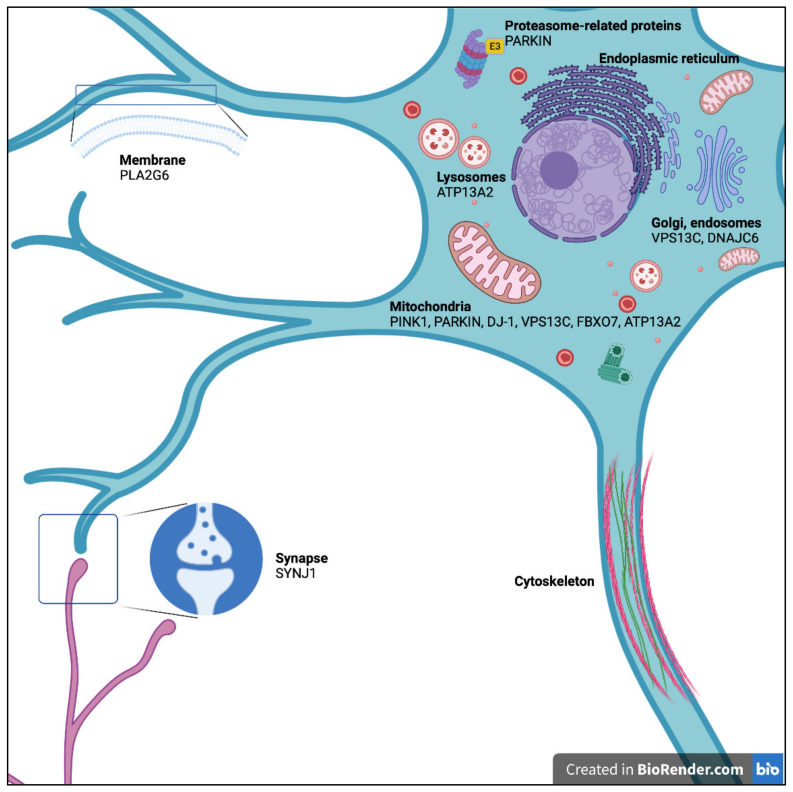
Subcellular localization of proteins encoded by genes involved in autosomal recessive Parkinson’s disease.

**Figure 2 biomedicines-09-00812-f002:**
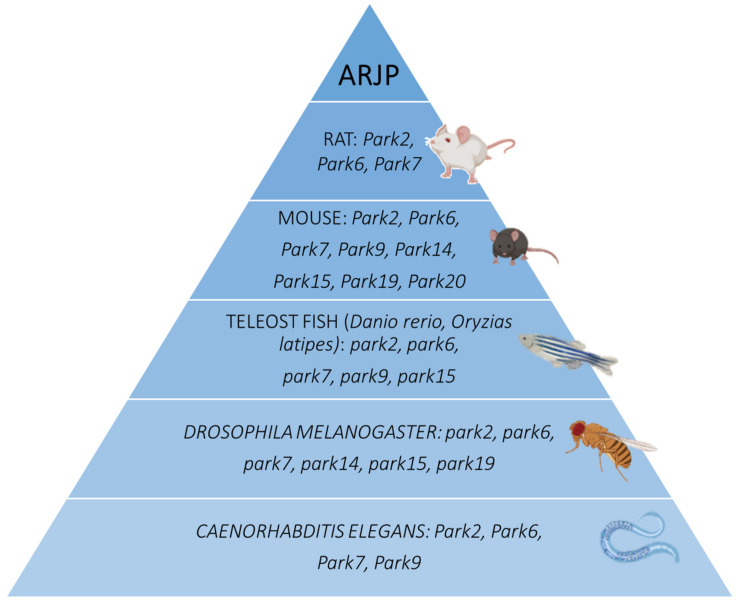
Animal models for the study of autosomal recessive Parkinson’s disease.

**Table 1 biomedicines-09-00812-t001:** Observed phenotype in animal models of *Parkin*-Linked Parkinson’s disease (*PARK2*).

Model	Phenotype	Mitochondrial Morphology	Mitochondrial Activity	Sensitivity to Oxidative Stress	Accumulation of Parkin SUBSTRATES	Reference(s)
**Exon 3-deleted *Park2* KO mouse (B6;129S2-Park2tm1Roo)**	Normal brain morphology, weight and size ⇩ body weight and temperature Absence of neurodegeneration No behavioral changes ⇩ DAT and VMAT2 striatal levels at 15 months of age Deficit in learning and memory	N.D.	N.D.	N.D.	N.D.	[22]
**Exon 3-deleted *Park2* KO mouse (B6;129S4-Park2tm1Shn)**	Normal brain morphology Absence of neurodegeneration ⇧ in extracellular DA concentration in the striatum ⇩ in synaptic excitability ⇩ behavioral performance in tests sensitive to nigrostriatal dysfunction	N.D.	⇩ levels of proteins involved in mitochondrial function	⇩ levels of proteins involved in oxidative stress	N.D.	[23,24]
**Exon 3-deleted *Park2* KO mouse (NeoR cassette)**	Absence of neurodegeneration in the SNcNo motor disabilities	Electron dense inclusion bodies in mitochondria Dilated and disorganized cristae	⇩ respiratory complex I activity in the SNc	N.D.	N.D.	[25]
**Exon 2-deleted *Park2* KO mouse (B6;129S4-Park2tm1Rpa)**	Absence of DA neuron degeneration No abnormalities in locomotor activities No changes in catecholamine levels	N.D.	N.D.	N.D.	N.D.	[26]
**Exon 2-deleted *Park2* KO mouse (MC1-NeoR + in-frame insertion of tauGFP fusion protein)**	No abnormalities in locomotor activities Absence of DA neuron degeneration No alteration in DAT level ⇩ release of DA	N.D.	N.D.	N.D.	N.D.	[27]
**Exon 7-deleted *Park2* KO mouse (B6;129S7/S4-Park2tm1Tmd)**	Absence of DA neuron degeneration Early loss of catecholaminergic neurons in LC ⇩ of norepinephrine-dependent acoustic startle response ⇩ of norepinephrine concentration in olfactory bulb and spinal cord	N.D.	N.D.	N.D.	N.D.	[28]
**Conditional exon 7-deleted *Park2* KO mouse using lentiviral delivery of GFP-tagged Cre-recombinase in midbrain of adult mice**	Progressive loss of DA neurons	⇩ in mitochondrial size, number and protein markers in ventral midbrain Defect in mitochondrial biogenesis	N.D.	N.D.	⇧ levels of PARIS	[29,30]
**Spontaneus mouse mutant quaking^viable^** **(qkv)**	Normal cellular conformation in the grey matter Alteration in DA metabolism ⇩ locomotor and exploratory activity Tremor in the caudal part of the trunk and proximal portions of the hind extremities Absence of DA neuron degeneration in the SNc Dysmyelination phenotype	N.D.	N.D.	N.D.		[31,32]
**BAC transgenic mouse expressing the C-terminal truncated human parkinQ311X mutation in DA neurons**	Multiple late onset and progressive hypokinetic motor deficits Age-dependent DA neuron degeneration in the SNc and loss of striatal DA neuron terminals ⇩ in striatal DA level	N.D.	N.D.	⇧ levels of nitrotyrosine	Age-dependent accumulation of proteinase K-resistant endogenous α-syn in the SNc	[33]
**Exon 4-deleted *Park2* KO rat**	Small reduction in DA neurons at 8 months of age Unaltered core body temperature ⇧ body weightNo neurochemical changes ⇩ activities of MAOs at 2 months of age ⇧ of β-PEA levels at 2 months of ageNormal behavior and locomotor activity in basal condition ⇩ locomotor activity upon low dose of methamphetamine	Mitochondrial pathological alteration	No mitochondrial dysfunctions	N.D.	N.D.	[34,35,36]
**Human *Park2* T240R overexpression mutant rat**	Progressive DA neuron death starting 8 weeks after rAAV2/8 injection	N.D.	N.D.	N.D.	N.D.	[37]
***park2* KO zebrafish**	⇩ in ascending DA neuron number in the posterior tuberculum	Normal mitochondrial morphology	Respiratory complex I deficiency	⇧ sensitivity to MPP^+^ ROS production	N.D.	[38]
***Parkin* KO *Drosophila***	Absence of neurodegeneration ⇩ lifespan Male sterility Severe disruption of muscle integrity Locomotion defects Myofibril degeneration	Swollen mitochondria with degenerated cristae	N.D.	N.D.	N.D.	[39,40]
***Parkin* KO *Drosophila* using P-element mutagenesis**	Absence of DA neuron degeneration Male and female infertility ⇩ lifespan Severe disruption of muscle integrity Locomotion defects Myofibril degeneration	N.D.	N.D.	⇧ sensitivity to chemical and environmental stress	N.D.	[41]
**Human ParkinR275W overexpression mutant *Drosophila***	Age-dependent degeneration of DA neuronal clusters Locomotor deficits that accelerate with age	Mitochondrial abnormalities in flight muscles		⇧ sensitivity to rotenone		[42]
**N-terminal deleted *parkin* and *parkin*K71P mutant *Drosophila***	⇩ longevity Drooped wing phenotype Locomotor dysfunction Muscle degeneration accompanied by apoptosis Severe loss of DA neurons Shrunken morphology of DA neurons Rescue of the phenotype by overexpression of WT *parkin*	N.D.	N.D.	N.D.	N.D.	[43]
**ParkinQ311X and *parkin* T240R mutant *Drosophila***	Age-dependent DA neuron degeneration ⇩ in climbing ability Severe motor deficits 2–3 weeks after eclosion	N.D.	N.D.	N.D.	N.D.	[44]
**PARKIN KO *C. elegans***	Normal development Shorter lifespan	N.D.	Mitochondrial complex I vulnerability	⇧ sensitivity to mitochondrial complex I inhibitors	N.D.	[45]
**PARK2-pdr1(gk448) III (CGC) mutant *C. elegans***	No Mn-induced degeneration of the CEP dopaminergic neurons	N.D.	N.D.	⇧ hypersensitivity to Mn-induced lethality Time-dependent ⇧ in Mn-induced RONS	N.D.	[46]

⇧ menas increase; ⇩ mean decrease.

**Table 2 biomedicines-09-00812-t002:** Observed phenotype in animal models of PTEN-induced kinase 1-linked Parkinson’s disease (PARK6).

Model	Phenotype	Mitochondrial Morphology	Mitochondrial Activity	Sensitivity to Oxidative Stress	Reference(s)
***Pink1* RNAi knockdown mouse**	Absence of DA neuron degeneration in the SNc No alteration in DA level in the striatum No abnormalities in spontaneous locomotor activities	N.D.	N.D.	N.D.	[67]
**Exon 4-7-deleted *Pink1* KO mouse**	Absence of DA neuron degeneration No alteration in striatal DA levels No alteration in DA synthesis or DA receptors levels No abnormalities in spontaneous locomotor activities ⇩ in evoked DA release in striatal slices ⇩ in the quantal size and release frequency of catecholamine in dissociated chromaffin cells ⇩ in corticostriatal LTP and LTD	No gross ultrastructural alterations No changes in the total number of mitochondria ⇧ number of larger mitochondria in the striatum at 3–4 and 24 months	Age-dependent impairment of mitochondrial function ⇩ in respiratory complex I and II activity in the striatum (young and old mice) and cerebral cortex (old mice) ⇩ in aconitase activity in the striatum (young and old mice) and cerebral cortex (old mice)	Mitochondria in the cortex are more sensitivity to oxidative stress	[68,69]
**Exon 2-3-deleted *Pink1* KO mouse**	Absence of DA neuron degeneration at 6 and 19 months of age No alteration in striatal DA content at 6 and 19 months of age No abnormalities in the spontaneous locomotor activity and normal motor coordination at 3 and 24 months Gait alterations and olfactory dysfunctions at 26 months ⇩ of the density of serotoninergic fibers in the glomerular layer of the olfactory bulb at 26 monthsAged males *Pink1* KO mice showed a significant deficit in the fine olfactory discrimination and in smell sensitivity	Less fragmented mitochondria	N.D.	N.D.	[70]
***Pink1* KO mouse (*Pink1^tm1Shn^*)**	Absence of DA neuron degeneration in the SNc No alteration in TH optical density in the striatum Vocalization’s impairments Impairment in limb motor skills with fewer hindlimb and forelimb steps ⇩ rearing and landing on the cylinder test Impairment during the pole test	N.D.	N.D.	N.D.	[71]
**G309D-*Pink1* transgenic mouse**	Age-dependent ⇩ of DA level ⇩ spontaneous locomotor activity Progressive ⇩ of body weight from middle age Absence of nigrostriatal degeneration No Lewy Bodies formation	⇩ of *Mtp18* Normal mitochondrial morphology and mass	⇩ ATP levels ⇩ mitochondrial membrane potential ⇩ respiratory complex activity	⇧ sensitivity to proteasomal stress	[72]
**Exon 2-5-deleted *Pink1* KO mouse (*in vitro* studies on primary neuronal cultures)**	Age-dependent ⇩ in long-term viability of cortical neuron cultures	N.D.	Mitochondrial calcium overload in primary cortical and midbrain neurons Loss of mitochondrial membrane potential ⇩ respiratory complex activity	⇧ cytotoxicity indices in cortical neuron cultures ⇧ ROS production in primary cortical and midbrain neurons	[65,73]
***Pink1* KO rat**	⇧ of striatal densities of DA D_2_ and D_3_ receptors at 6 months of age No changes in striatal density of DA D_1_ receptors, VMAT2 and DAT at 6 months of ageMotor impairments in movement, strength and coordination starting from 4 months of age Loss of SNc DA neurons starting from 6 months No change in striatal TH or α-syn immunoreactivity within the SNc and striatum ⇧ of DA and 5-HT striatal content at 8 months of age ⇧ glycolysis in the striatum Early and progressive vocalization impairment and oromotor deficits Compromised communication Sensorimotor deficits with a ⇩ in spontaneous activity starting from 8 months No change in TH immunoreactivity in the striatum and in the SNc at 8 months of age ⇩ in TH immunoreactivity in LC at 8 months of age α-syn aggregates in PAG, SNc and LC at 8 months	⇧ DRP1 and ⇩ MFN2 in the striatum at 4 months of age ⇧ mitochondrial fission and fragmentation	⇩ in the level of respiratory complex I, III, IV, V in the striatum ⇧ proton leak at 4 and 9 months of age	⇧ ROS generation Altered stress pathway in the striatum at 4 months of age	[34,74,75,76,77]
**Zebrafish MO-mediated *pink1* knockdown**	Structural alterations in the axonal scaffold ⇩ number of central DA neurons Lower heart rate ⇧ of VEGF and erythropoiesis ⇩ *th1* and *th2* mRNA, but normal levels of *dat* mRNA Locomotor dysfunctions Weak or absent response to tactile stimuli ⇩ swimming behavior ⇧ susceptibility to MPTP-induced motor disturbances	N.D.	⇩ mitochondrial membrane potential ⇧ GSK-3β activity Alteration of mitochondria biogenesis	⇧ ROS levels ⇧ oxidative stress ⇧ caspase-3 activity ⇩ *hif1α* mRNA level ⇩ catalase enzyme activity ⇩ *catalase* and *SOD3* transcript ⇧ susceptibility to MPTP ⇧ susceptibility to H_2_O_2_	[8,78,79,80]
**Y431* *pink1* transgenic zebrafish**	Progressive loss of DA neurons from 5 dpf to 18 months of age No obvious behavioral abnormalities Microglial activation	Enlarged mitochondria	⇩ mitochondrial complex I and III activity ⇧ *TigarB* expression	N.D.	[55,81]
**Q178X *pink1* transgenic Medaka**	Late-onset motor deficits ⇩ in the frequency of spontaneous swimming movements Shortened lifespan Absence of neurodegeneration ⇩ DOPAC	Normal mitochondrial morphology	N.D.	N.D.	[82]
***pink1* KO *Drosophila***	Male sterility Degeneration of flight muscles Mild loss of DA neurons Impaired flight abilityS evere ⇩ climbing rate Abnormally positioned wings Crushed thorax Disorganized muscle fibers Muscle cell apoptosis Impaired mobilization of synaptic vesicle reserve pool during rapid stimulation Synaptic ATP depletion	Enlarged and swollen mitochondria with loss of the outer membrane Fragmented cristae	⇩ ATP levels and synthesis ⇩ mitochondrial complex I and IV activity Deficit in mitochondrial membrane potential	⇧ oxidative stress	[83,84,85,86]
***pink1* RNAi knockdown *Drosophila***	⇩ lifespan Abnormal wing posture Disruption of muscle integrity Degeneration of indirect flight muscles Impaired flight ability (limited to the early days of life) ⇩ in climbing ability Degeneration of TH-positive neurons ⇩ of brain DA content	Grossly swollen mitochondria lacking electron-dense material Disintegrated cristae	⇩ ATP levels	N.D.	[87]
***C. elegans pink*-*1(tm1779)* mutant**	Defects in axonal outgrowth of CAN ⇩ of lifespan	⇩ in mitochondrial cristae length ⇧ fused mitochondrial network Fragmented mitochondria	No difference in basal OCR ⇧ OCR after FCCP exposure ⇩ OCR after DCCD exposure ⇧ proton leak ⇩ ATP levels ⇩ Mitochondrial membrane potential Loss of mitophagy ⇩ in mitochondrial turnover	⇧ paraquat sensitivity ⇩ oxidative stress response ⇧ DCCD and FCCP sensitivity ⇧ mitochondrial ROS production	[88,89,90,91,92]

⇧ menas increase; ⇩ mean decrease.

**Table 3 biomedicines-09-00812-t003:** Observed phenotypes in animal models of *PARK7*-linked Parkinson’s disease.

Model	Phenotype	Sensitivity to Oxidative Stress	Mitochondrial Activity	Reference(s)
**Exons 3-5-deleted *Dj-1* KO mouse**	Absence of DA neuron degeneration No motor deficits observed by pole test, open field, adhesive removal test No changes in the density of striatal TH fibers and DAT level No alteration in DA striatal levels Motor deficit observed on running wheels and rotarod	⇧ susceptibility to MPTP-induced neuron loss	N.D.	[106,107]
**Exons 1-5-deletd *Dj-1* KO mouse**	Absence of DA neuron degeneration No motor deficits observed by rotarod test at any age Motor impairments at 11 months of age with open field test Nigrostriatal dysfunction with tape removal task at 5 and 11 months of age ⇧ evoked release of DA in dorsal striatum Age-dependent ⇧ in striatal DA content No differences in TH, DAT and VMAT2 protein levels No α-syn- or ubiquitin-positive inclusions in the SNc	N.D.	N.D.	[103]
**Exon 2-deleted *Dj-1* KO mouse**	Absence of DA neuron degeneration No alteration in DA levels in basal ganglia Normal TH activity No α-syn- or ubiquitin-positive inclusions in the SNc ⇩ evoked release of DA with amperometry analysis Normal LTP induction, but absence of LTD Motor impairments at 3 months of age with open field, rotarod and startle tests	N.D.	N.D.	[99]
***Dj-1* mouse with deletion of intron between exons 6 and 7**	No change in the number of TH-positive and Nissl-stained nigral cells No change in DA, DOPAC, HVA No behavioral impairment Alteration in striatal DA transmission ⇧ in striatal DAT in synaptosomal fraction	N.D.	N.D.	[108]
***Dj-1* KO rats**	Loss of DA neurons in the SNc and LC at 8 months No change in TH immunoreactivity in the striatum ⇧ in striatal DA and 5-HT content at 8 months of age No change in DAT density in striatum ⇧ in VMAT2 and D_1_, D_2_ and D_3_ DA receptor density in striatum between 4 and 8 months of age ⇧ in body weight Motor impairments in movement, strength and coordination between 6 and 8 months Deficit in cylinder test from 4 to 13 months of age No alterations in sensorimotor functions with adhesive removal test No anxiety or depression at 4, 8 or 17 months of age Abnormality in the neuroendocrine system	N.D.	N.D.	[34,74,109,110,111]
**Zebrafish MO-mediated *dj-1* KO**	No loss of DA neurons in basal condition Loss of DA neurons after exposure to H_2_O_2_ or to proteasome inhibitor MG132⇧ levels of p53 and Bax	⇧ susceptibility to H_2_O_2_	N.D.	[112,113]
**CRISPR-Cas9 *dj-1* KO Zebrafish**	Smaller size and ⇩ in body mass starting from 3 months of age ⇩ in TH levels and DA content at 16 months of age Locomotor deficits (bradykinesia): reduction in distance travelled, velocity, time spent moving and duration of a swimming episode	N.D.	⇩ mitochondrial complex I activity in skeletal muscle at 16 months of age	[114,115]
***dj-1β* KO *Drosophila***	No loss of DA neurons ⇩ taste sensitivity Defective in ability to form associative memories ⇩ climbing ability and further loss of climbing activity after repeated paraquat exposure	⇧ susceptibility to H_2_O_2_ Resistance to oxidative stress induced by paraquat	N.D.	[116,117,118]
***dj-1α* and *dj-1β* double KO *Drosophila***	No loss of DA neurons Normal lifespan	⇧ sensitivity to H_2_O_2_, paraquat and rotenone	N.D.	[119]
***djr-1.1* KO *C. elegans***	N.D.	⇧ susceptibility to rotenone	N.D.	[45]
***djr-1.1* and djr-1.2 double KO *C. elegans***	⇧ inflammatory signaling after exposure to *Pseudomonas aeruginosa*	N.D.	N.D.	[120]
***djr* KO *C. elegans***	⇩ survival and lifespan after acute Mn exposure in *djr-1.2* or *djr* double deletion mutants ⇧ dauer movement in *djr-1.2* deletion mutant	⇧ sensitivity to Mn in *djr-1.2* or *djr* double deletion mutants	N.D.	[121]

⇧ menas increase; ⇩ mean decrease.

**Table 4 biomedicines-09-00812-t004:** Observed phenotype in animal models of *PARK9*-linked Parkinson’s disease (*ATP13A2*).

Model	Phenotype	Sensitivity to Oxidative Stress	Mitochondrial Activity	Reference(s)
***Atp13a2* KO mouse**	Late onset sensorimotor and cognitive deficit (20–29 months of age) Late onset motor impairments (20–29 months of age) No loss of DA neurons in the SNc No change in striatal DA levels Gliosis at 1 month of age Lipofuscinosis at 3 months of age Accumulation of LAMP1, LAMP2 and BMP at 6 months of age Aggregation of ubiquitinated proteins and p62 at 12 months of age Aberrant processing of the lysosomal protease CATD at 12 months of age Accumulation of insoluble α-syn in the hippocampus at 18–20 months of age	N.D.	N.D.	[131,135,136,137]
**Zebrafish MO-mediated *atp13a2* knockdown**	Complete abrogation of *atp13a2* led to embryonic lethality Partial abrogation of *atp13a2* allowed offspring survival Curved phenotype at 48 hpf Movement latency and abnormal response to environmental stimulus at 7 dpf	N.D.	N.D.	[138]
**Zebrafish *atp13a2^sa18624^* and *atp13a2^sa14250^* mutant**	No obvious morphological or behavioral abnormality At 5 dpf, pericardial oedemas, movement loss, spine curvature and underdevelopment of the swimming bladder in homozygous *atp13a2* mutants exposed to Mn^2+^ Apoptotic areas throughout the CNS after exposure to Mn^2+^ in *atp13a2^sa18624^*^−/−^ larvae	⇧ susceptibility to Mn^2+^	N.D.	[139]
**CRISPR-Cas9 KO *atp13a2* Zebrafish**	Loss of DA neurons in posterior tuberculum and norepinephrine neurons in LC at both 4 and 12 months of age CATD deficiency Lysosomal and intracellular vesicle trafficking dysfunction	N.D.	N.D.	[140]
***atp13a2* “IVS13, T-C, +2” mutant medaka**	Shorter lifespan ⇧ spontaneous swimming movement at 4 months, but no differences in swimming at 12 months Age-dependent and progressive loss of DA neuron in the middle diencephalon at 8 and 12 months of age ⇩ density of TH-positive fibers in the telencephalon at 8 and 12 months ⇩ of noradrenergic neurons in the medulla oblongata at 8 and 12 months of age ⇩ in DA content at 12 months of age ⇩ CATD protein level and activity Fingerprint-like subcellular structures in the brain	N.D.	N.D.	[141]
***catp-6(ok3473) C. elegans* mutant**	⇩ locomotion Delay in the rate of development Higher mortality rate in midlife age Alteration in iron homeostasis Down-regulation of the core genes required for metabolizing iron Altered Zn homeostasis after Zn exposure ⇩ of cleaved LGG1/LC3-II protein levels ⇧ P-CATD levels ⇩ mRNA levels of genes required for autophagy and lysosomal function	N.D.	⇩ of mitochondrial membrane potential ⇩ of maximal respiration rate ⇧ sensitivity to rotenone	[142,143]

**Table 5 biomedicines-09-00812-t005:** Observed phenotypes in animal models of *PARK14*-linked Parkinson’s disease (*PLA2G6*).

Model	Phenotype	Mitochondrial Activity	Sensitivity to Oxidative Stress	Reference(s)
***Pla2g6* KO mouse**	Degeneration of nigrostriatal DA neurons and loss of striatal TH and nerve terminal DAT from 56 weeks of age Axonal degeneration and atrophic axonsLate onset motor dysfunctions (2 years of age) Presence of spheroids and vacuoles throughout the CNS and the PNS ⇧ expression of α-syn and phosphorylated α-syn in mitochondria ⇩ male fertility ⇩ in DHA metabolism at 4 months of age	N.D.	N.D.	[152,153,154,155]
**Exon2-deleted *Pla2g6* KO mouse (*Pla2g6* ex2^KO^)**	Progressive loss of SNc DA neurons starting from 16 months of age Progressive age-dependent L-DOPA-sensitive motor dysfunctions starting from 16 months of age Impairment of PLA2G6-dependent Ca^2+^ signaling and depletion of intracellular Ca^2+^ stores in MEFs Accumulation of LC3 and increased autophagosome numbers in the SNc DA neurons	N.D.	N.D.	[156]
**D331Y KI *Pla2g6* mouse**	Loss of DA neurons in the SNc at 6 and 9 months Degeneration of nigrostriatal dopaminergic terminals at 9 months Presence of Lewy Bodies in the SNc at 9 months ⇧ α-syn and phosphorylated α-syn expression in the SNc at 9 months Early-onset and progressive PD phenotype from 6 to 12 months: slowness of movement, hypoactivity, impaired motor coordination and performance	Disrupted structure of mitochondria cristae ⇩ of mitochondrial size ⇩ mitochondrial complex I and III activity ⇩ ATP level ⇧ cytosolic level of cytochrome c Activation of mitochondrial apoptotic pathway Mitophagy impairment	⇧ endoplasmic reticulum stress ⇧ ROS production in the SNc ⇧ lipid peroxidation	[157]
***pla2g6* KO *Drosophila* (iPLA2-VIA KO)**	Progressive degeneration of DA neurons Impaired synaptic transmission ⇧ ceramide production Shorter lifespan α-syn aggregates	N.D.	N.D.	[146,158]

⇧ menas increase; ⇩ mean decrease.

**Table 6 biomedicines-09-00812-t006:** Observed phenotypes in animal models of *PARK15*-linked Parkinson’s disease (*FBXO7*).

Model	Phenotype	Mitochondrial Activity	Reference(s)
**Exon4-deleted *Fbx07* KO mouse**	No difference in body or brain weight at P5 Lower body and brain weight at P18 Early-onset motor deficits at P18 Premature death in the 4th postnatal week Moderate ⇧ in cell death in the cortex at P18 Absence of DA neuron degeneration in the SNc at P18 No change in the levels of DA and its metabolites in the striatum at P18 Absence of α-syn protein deposits at P18 Astrogliosis in the cortex at P18	N.D.	[167]
***Fbx07* conditional KO mouse (Nex-Cre;fl/fl). Deletion of *Fbx07* from pyramidal neurons of the cortex and hippocampus**	Longer lifespan (at least up to 4 months) Spasticity and progressive motor coordination deficits at 2 and 4 months of age Normal body weight at 2 months of age, with successive stagnation Astrogliosis in the cortex	N.D.	[167]
***Fbx07* conditional KO mouse (TH-Cre;fl/fl). Deletion of *Fbx07* gene from TH-expressing neurons**	Longer lifespan (at least up to 12 months) Absence of motor deficit at 2 months of age Progressive motor deficit at 6 and 12 months of age: slow movements, reduced mobility, alteration in several fine gait parameters Absence of DA neuron degeneration at 2 and 12 months of age ⇩ of DA content (50%) in the striatum at 2 and 12 months ⇧ of astrogliosis at 12 months in the SNc	N.D.	[167]
**Zebrafish MO-mediated *zFbxo7* knockdown**	Knockdown of zFbxo7 protein expression Mild and severe morphological phenotype correlated with the silencing of zFbxo7 protein at 72 hpf: curly tails, heart edema, heart malformations Locomotor impairments in the ATG-MO-Mild and SP-MO-Mild morphants ⇩ of number of DA neurons in SP-MO-severe morphants ⇩ of number of DAT-positive neurons in ATG-MO and SP-MO morphants	N.D.	[168]
**hFBXO7 WT overexpression *Drosophila* mutants**	Locomotor deficits in flight and climbing ability starting from 30–40 days Loss of DA neurons in PPM1/2 and PPM3 DA neuron cluster at 40 days FBXO7 protein aggregation	Mitochondrial deficit Swelling of muscle mitochondria Broken mitochondria cristae Accumulation of high-density materials in the swollen mitochondria	[166]

⇧ menas increase; ⇩ mean decrease.

**Table 7 biomedicines-09-00812-t007:** Observed phenotypes in animal models of *PARK19*-linked Parkinson’s disease (*DNAJC6*).

Model	Phenotype	Mitochondrial Morphology	Sensitivity to Oxidative Stress	Reference(s)
***Dnajc6* KO mouse**	High rate of early postnatal mortality, with an apparently normal lifespan of the surviving pups ⇩ body weight Delayed female sexual maturity ⇧ GAK levels in the brain in embryos and in 3–5-week-old mice	N.D.	N.D.	[177]
***dnajc6* RNAi knockdown *Drosophila***	⇩ lifespan Age-dependent ⇩ of DA neuron number Locomotor deficit in climbing ability	No alterations	⇧ sensitivity to paraquat-induced oxidative stress	[178]

⇧ menas increase; ⇩ mean decrease.

**Table 8 biomedicines-09-00812-t008:** Observed phenotype in animal models of *PARK20*-linked Parkinson’s disease (*SYNJ1*).

Model	Phenotype	Reference(s)
***Synj1* KO mouse**	Severe reduction of milk in their stomachs few hours after birth Higher post-natal mortality rate (85% within 24 h and 15% within 15 days) Severe weakness, ataxia and generalized convulsions on stimulation by nociceptive stimuli at 10 days of age ⇩ body weight at 10 days of age ⇧ number of clathrin-coated vesicles in the cytomatrix-rich area Impairment in synaptic vesicle recycling ⇩ of synaptic transmission after prolonged high-frequency stimulation and delayed recovery after interruption of the stimulus in CA1 area	[185]
***Synj1* heterozygous mouse**	Normal body weight Normal lifespan Age-dependent motor impairments ⇩ of striatal DA, DOPAC and HVA content at 7 and 12 months Progressive ⇩ of DA terminals from 3 to 18 months Accumulation of pS129 α-syn in the cortex, striatum and midbrain at 18 months	[186]

⇧ menas increase; ⇩ mean decrease.

## Data Availability

Not applicable.

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
