# Peer review of "Animal Models of Autosomal Recessive Parkinsonism"

_biomedicines, 2021, doi:10.3390/biomedicines9070812_

Round 1

Reviewer 1 Report

The authors reviewed literature data to address the use of animal models of autosomal recessive parkinsonism to explains the motor and non-motor symptoms in PD. Recently many reviews have been published on the animal model of PD. Authors should revise the manuscripts to make the following changes for improving paper quality.

  • Please add details of neuronal loss, L-DOPA responsiveness, Lewy Body –Like Inclusions, and suitability for testing disease for each animal model in the table.
  • Add an illustrative diagram that shows the mechanism of autosomal recessive in PD.
  • Please add a summarizing figure that supports the conclusions section.

Author Response

The authors reviewed literature data to address the use of animal models of autosomal recessive parkinsonism to explains the motor and non-motor symptoms in PD. Recently many reviews have been published on the animal model of PD. Authors should revise the manuscripts to make the following changes for improving paper quality.

Comments:

  • Please add details of neuronal loss, L-DOPA responsiveness, Lewy Body –Like Inclusions, and suitability for testing disease for each animal model in the table.

We thank the reviewer for this suggestion. The tables show the characteristics described in the main text and figures for each animal model. Unfortunately, not all the studies investigated neuronal loss, L-DOPA responsiveness, the presence of Lewy Body –Like Inclusions, and suitability for testing disease. Thus, these data  for certain animal models are not available in the  literature and could not be included in the review.

In particular, in table 1, under the column “Phenotype” we reported  the presence or absence of neurodegeneration in all models, except for PARKIN KO C. elegans (data not available in the original paper). Under the column “Mitochondrial morphology” only in the model Exon 3-deleted Park2 KO mouse (NeoR cassette) it was possible to report the presence of inclusion bodies in mitochondria. Other details such as L-DOPA responsiveness were not investigated in the original studies on PARK2 genetic models.

In table 2, under the column “Phenotype” in all models, except for the Exon 2-5-deleted Pink1 KO mouse and the C. elegans pink-1(tm1779) mutant, it was reported the presence or absence of neurodegeneration. We also reported in G309D-Pink1 transgenic mouse the absence of Lewy Body, and in Pink1 KO rat no α-syn aggregates. Other details were not reported in the original studies on PARK6 models.

In table 3, under the column “Phenotype” in all models, except for CRISPR-Cas9 dj-1 KO Zebrafish, djr-1.1 KO C. elegans, djr-1.1 and djr-1.2 double KO C. elegans and djr KO C. elegans, we reported the presence or absence of neurodegeneration. Furthermore, in the models exons 1-5-deletd Dj-1 KO mouse and exon 2-deleted Dj-1 KO mouse it was reported no α-syn- or ubiquitin-positive inclusions. Other details were not reported in the original studies on PARK7 models.

In table 4, under the column “Phenotype” in the models Atp13a2 KO mouse, CRISPR-Cas9 KO atp13a2 Zebrafish, and atp13a2 “IVS13, T-C, +2” mutant medaka it was reported the presence or absence of neurodegeneration, and in Atp13a2 KO mouse it was reported the accumulation of insoluble α-syn. Other details were not reported in the original studies on PARK9 models.

In table 5, under the column “Phenotype” in all animal models it was reported the presence or absence of neurodegeneration. In the Pla2g6 KO mouse, in D331Y KI Pla2g6 mouse and pla2g6 KO Drosophila (iPLA2-VIA KO) it was reported the presence of α-syn or Lewy Body. In Exon2-deleted Pla2g6 KO mouse (Pla2g6 ex2KO) it was described progressive age-dependent L-DOPA-sensitive motor dysfunctions. Other details were not reported in the original studies on PARK14 models.

In table 6, under the column “Phenotype” in all animal models, except for the Fbx07 conditional KO mouse (Nex-Cre;fl/fl), it  was reported the presence or absence of neurodegeneration. In Exon4-deleted Fbx07 KO mouse α-syn protein aggregates were not detected. Other details were not reported in the original studies on PARK15 models.

In table 7, under column “Phenotype” we reported the loss of DA neuron only in dnajc6 RNAi knockdown Drosophila.  Other details were not reported in the original studies on PARK19 models.

In table 8, under column “Phenotype” in Synj1 heterozygous mice we reported the progressive loss of DA terminals and accumulation of pS129 α-syn in the cortex, striatum and midbrain. In Synj1 KO mouse these features were not investigated.

Add an illustrative diagram that shows the mechanism of autosomal recessive in PD.

We thank the reviewer for this suggestion. As requested, we added a new figure (figure 1 of the revised version) to show the sites of action of the proteins encoded by genes responsible for autosomal recessive PD.

  • Please add a summarizing figure that supports the conclusions section.

As requested, we added a new figure (figure 2 of the revised version) to summarize all the animal models used for modeling autosomal recessive PD.

The text has been checked for grammar and style by a professional translator and native English speaker.

Reviewer 2 Report

"Animal models of autosomal recessive parkinsonism."

The authors reviewed animal models for research into the mechanism of Parkinson's Disease. Animal models used to study PD have been carefully presented, detailing the genetic basis of the changes. It is especially valuable to include in a table the most important points characterizing a given model and a detailed discussion of these changes in specific species.

The presented article may be a very good guide for researchers who, while working on changes in nervous tissue in the course of Parkinson's Disease, must choose the appropriate animal model. The article is written in an interesting way, detailing the strengths and weaknesses of each animal model presented and a brief history of its development.

The article is worth recommending.

Author Response

Referee#2

The authors reviewed animal models for research into the mechanism of Parkinson's Disease. Animal models used to study PD have been carefully presented, detailing the genetic basis of the changes. It is especially valuable to include in a table the most important points characterizing a given model and a detailed discussion of these changes in specific species.

The presented article may be a very good guide for researchers who, while working on changes in nervous tissue in the course of Parkinson's Disease, must choose the appropriate animal model. The article is written in an interesting way, detailing the strengths and weaknesses of each animal model presented and a brief history of its development.

The article is worth recommending.

We thank Reviewer #2 for the positive evaluation of our manuscript. We really appreciated your comments.